# An improved multivariable integrated evaluation method and NCL code for multimodel intercomparison (MVIETool version 1.0)

Meng-Zhuo Zhang[1,2], Zhongfeng Xu[2,3], Ying Han[2], Weidong Guo[1,3]

[1] Institute for Climate and Global Change Research, School of Atmospheric Sciences, Nanjing University, Nanjing, China
[2] CAS Key Laboratory of Regional Climate and Environment for Temperate East Asia, Institute of Atmospheric Physics, Chinese Academy of Sciences, Beijing 100029, China
[3] Jiangsu Collaborative Innovation Center for Climate Change, Nanjing University, Nanjing, China

*Correspondence to*: Zhongfeng Xu (xuzhf@tea.ac.cn)

**Abstract.** An evaluation of a model's overall performance in simulating multiple fields is fundamental to model intercomparison and development. A multivariable integrated evaluation (MVIE) method was proposed previously based on a vector field evaluation (VFE) diagram, which can provide quantitative and comprehensive evaluation on multiple fields. In this study, we make further improvements to this method from the following aspects. (1) We take area weighting into account in the definition of statistics in the VFE diagram and MVIE method, which is particularly important for a global evaluation. (2) We consider the combination of multiple scalar fields and vector fields against multiple scalar fields alone in the previous MVIE method. (3) A multivariable integrated skill score (MISS) is proposed as a flexible index to measure a model's ability to simulate multiple fields. Compared with the MIEI proposed in the previous study, MISS is a normalized index that can adjust the relative importance of different aspects of model performance. (4) A simple-to-use and straightforward tool, the Multivariable Integrated Evaluation Tool (MVIETool version 1.0), is developed to facilitate an intercomparison of the performance of various models. Users can use the tool coded either with the open-source NCAR Command Language (NCL) or Python3 to calculate the MVIE statistics and plotting. With the support of this tool, one can easily evaluate model performance in terms of each individual variable and/or multiple variables.

## 1 Introduction

An increasing number of model intercomparison projects (MIPs) have been carried out over the past decade (Eyring et al., 2016; Simpkins, 2017). The Coupled Model Intercomparison Project Phase 6 (CMIP6) includes more than 20 MIPs: e.g., the Radiative Forcing MIP (RFMIP), the Geoengineering MIP (GeoMIP), and the Global Monsoons MIP (GMMIP) (Kravitz et al., 2011; Pincus et al., 2016; Zhou et al., 2016). Quantitative evaluation and intercomparison of climate models have become increasingly important (Knutti et al., 2013) and have escalated the need for innovative and comprehensive approaches to model evaluation (Meehl et al., 2014; Stouffer et al., 2016; Eyring et al., 2019).

Climate models are commonly evaluated in terms of their ability to simulate historical climate compared to observed or reanalyzed data, using performance metrics (Pincus et al., 2008; Flato et al., 2013). A set of useful metrics and diagrams has

been developed for model evaluation. The widely-used Taylor diagram summarizes model performance in simulating a scalar field using correlation coefficient (CORR), standard deviation (SD), and root-mean-square difference (Taylor, 2001). Objective performance metrics (e.g., relative error and portrait diagrams) have been proposed for the evaluation of various variables (Glecker et al., 2008). Xu et al. (2016) devised a vector field evaluation (VFE) diagram which can be regarded as a generalized Taylor diagram. The VFE method allows an evaluation of a model's ability to simulate a vector field (Huang et al., 2019; 2020). Based on the VFE diagram, Xu et al. (2017) further developed a multivariable integrated evaluation (MVIE) method to evaluate model performance in terms of multiple fields by grouping various normalized scalar fields into an integrated vector field. The MVIE method also defined a multivariable integrated evaluation index (MIEI) to summarize the model's overall performance in simulating multiple fields. The MIEI, the VFE diagram, and the performance metrics of individual scalar variables constitute a hierarchical model evaluation framework, which can provide a quantitative and comprehensive evaluation of model performance.

However, the MVIE method proposed by Xu et al. (2017) considered only the integrated evaluation of various scalar fields. Under certain circumstances, both scalar variables and vector variables (e.g., air temperature and vector wind fields) warrant evaluation together. Moreover, the vector field statistics employed in Xu et al., (2016; 2017) did not consider area weighting, which is a limitation especially for an evaluation of the global field. Although area weighting was considered in many previous statistical metrics, e.g., correlation coefficient and standard deviation, they were used to evaluate scalar fields rather than vector fields (e.g., Watterson, 1996; Boer and Lambert, 2001; Masson and Knutti, 2011). The consideration of area weighting in the definition of vector field statistics is one of the novelty of our study relative to previous studies (Taylor, 2001; Boer and Lambert, 2001; Gleckler et al., 2008; Xu et al., 2016; 2017). In addition, we also improve MVIE method to allow a mixed evaluation of scalar and vector fields. Furthermore, based on MIEI, a multivariable integrated skill score (MISS) for a climate model is proposed, which allows us to adjust the relative importance of different aspects of model performance. Finally, we develop a Multivariable Integrated Evaluation Tool (MVIETool version 1.0) to facilitate multimodel intercomparison. These efforts are expected to improve the accuracy and flexibility of the VFE and MVIE methods.

The paper is organized as follows. Section 2 defines statistical metrics that take area weighting into account. Section 3 introduces the improved MVIE method with a combination of scalar and vector fields, and interprets the performance metrics. Section 4 gives an overview of the MVIETool and describes the technical process of the MVIETool, including setting of the arguments in scripts. In Section 5, the applications of the tool are demonstrated by showing three examples with ten CMIP Phase 5 (CMIP5) models. Finally, a summary is given in Section 6.

## 2 Statistical metrics

The MVIE method primarily consists of three statistical quantities — root-mean-square length (RMSL), vector similarity coefficient (VSC), and root-mean-square vector difference (RMSVD) — that measure model performance in simulating a vector field from various aspects (Xu et al., 2017). RMSL measures the magnitude of a vector field, VSC measures the

similarity of two vector fields, and RMSVD measures the overall difference between two vector fields. MIEI was defined by using root-mean-square (rms) values of all variables and VSC, and is a concise metric to rank models in terms of their performance in simulating multiple fields. However, the definition of these statistical quantities did not consider area weighting, which could to a certain extent misrepresent the relative contribution of different latitudes to the statistics. Here, we redefine these statistical quantities by taking area weighting into account.

Assume that there are $M$ variables derived from model A and observation O. We need to normalize each modeled variable using the rms value of the corresponding observed variable. The normalized $M$ variables are dimensionless and can be grouped into $M$-dimensional vector fields for model $A$ and observation $O$:

$$\boldsymbol{A_j} = \left(a_{1j}, a_{2j}, \ldots, a_{Mj}\right); \quad j = 1, 2, \ldots, N$$

$$\boldsymbol{O_j} = \left(o_{1j}, o_{2j}, \ldots, o_{Mj}\right); \quad j = 1, 2, \ldots, N$$

Each field is composed of $N$ vectors in time and/or space and $M$ is the dimension of the integrated vector field.

## 2.1 Uncentered statistics

Similar to the weighted statistics defined by Watterson (1996), we define the weighted RMSL ($L_A$, $L_O$), VSC ($R_v$), and RMSVD as follows:

$$L_A = \sqrt{\sum_{i=1}^{M} \sum_{j=1}^{N} w_j a_{ij}^2}, L_O = \sqrt{\sum_{i=1}^{M} \sum_{j=1}^{N} w_j o_{ij}^2} \tag{1}$$

$$R_v = \frac{\sum_{i=1}^{M} \sum_{j=1}^{N} w_j a_{ij} o_{ij}}{L_A \cdot L_O} \tag{2}$$

$$\text{RMSVD} = \sqrt{\sum_{i=1}^{M} \sum_{j=1}^{N} w_j (a_{ij} - o_{ij})^2} \tag{3}$$

where $w_j$ is the area weighting factor and the sum of $w_j$ is equal to 1. In terms of equal weight, $w_j$ is equal to $1/N$ for all $j$. These uncentered metrics focus mainly on different aspects of the vector field. With the aid of Eq. (3), the square of RMSVD can be written as:

$$\text{RMSVD}^2 = \sum_{i=1}^{M} \sum_{j=1}^{N} w_j \left(a_{ij} - o_{ij}\right)^2$$

$$= \sum_{i=1}^{M} \sum_{j=1}^{N} w_j \left(a_{ij}^2 + o_{ij}^2 - 2 \cdot a_{ij} \cdot o_{ij}\right)$$

$$= \sum_{i=1}^{M} \sum_{j=1}^{N} w_j a_{ij}^2 + \sum_{i=1}^{M} \sum_{j=1}^{N} w_j o_{ij}^2 - 2 \cdot \sum_{i=1}^{M} \sum_{j=1}^{N} w_j a_{ij} o_{ij} \tag{4}$$

With the aid of Eqs. (1)–(3), Eq. (4) can be written as:

$$\text{RMSVD}^2 = L_A^2 + L_O^2 - 2 \cdot L_A \cdot L_O \cdot R_v \tag{5}$$

Note that RMSVD, $L_A$, $L_O$, and $R_v$ with area weighting still satisfy the cosine law (Eq. 5). Thus, the VFE diagram is still valid with these weighted statistics (Eq. 5). We define the standard deviation of rms values (rms_std) to quantify the dispersion of the rms values of $M$ variables:

$$\sigma_{rms} = \sqrt{\frac{1}{M}\sum_{m=1}^{M}\left(L_{Am}^* - \frac{1}{M}\sum_{m=1}^{M}L_{Am}^*\right)^2} \qquad (6)$$

where $L_{Am}^* = \frac{L_{Am}}{L_{Om}}$ is the ratio of the modelled rms value of the $m$th component (variable) to the observed rms value.

Note that RMSVD does not decrease monotonically with an improvement in model performance. To measure model performance more accurately, Xu et al. (2017) devised a multivariable integrated evaluation index, termed MIEI, of climate model performance:

$$\text{MIEI}^2 = \frac{1}{M}\sum_{m=1}^{M}(L_{Am}^* - 1)^2 + 2 \cdot (1 - VSC) \qquad (7a)$$

Note that the first and second terms of the right-hand side of Eq. (7a) can vary from 0 to $+\infty$ and from 0 to 4, respectively. Thus, the MIEI may be too sensitive to rms bias and insensitive to pattern bias. To fix this problem we redefine MIEI as follows:

$$\text{MIEI}^2 = \frac{1}{M}\sum_{m=1}^{M}(R_m^* - 1)^2 + 2 \cdot (1 - VSC) \qquad (7b)$$

where $R_m^*$ is defined as:

$$R_m^* = \begin{cases} L_{Am}^*, & L_{Am}^* \leq 1 \\ \frac{1}{L_{Am}^*}, & L_{Am}^* > 1 \end{cases} \qquad (8)$$

$R_m^*$ varies from 0 to 1. Here, we assume that $L_{Am}^*$ and $\frac{1}{L_{Am}^*}$ represent the same model performance except that one overestimates rms and the other underestimates rms. MIEI takes the rms values and VSC into consideration at the same time.

The relative importance of a model's ability to simulate the pattern similarity and amplitude of variables depends on the application. Hence, a weight factor $F$ is added to the MIEI to adjust the relative importance of rms and VSC:

$$\text{MIEI}^2 = \frac{1}{M}\sum_{m=1}^{M}(R_m^* - 1)^2 + F \cdot (1 - VSC) \qquad (9)$$

We can further define a multivariable integrated skill score (MISS) of a climate model:

$$\text{MISS} = (F + 1 - \text{MIEI}^2)/(F + 1) \qquad (10)$$

MISS varies from $-F/(F+1)$ to 1. MISS reaches its minimum value of $-F/(F+1)$ when VSC equals $-1$ and $R_m^*$ equals 0. Note that VSC is usually greater than 0 in terms of model evaluation. Thus, we find that MISS usually varies from 0 to 1. It is very unlikely that MISS will be less than 0, unless VSC is less than 0. MISS varies monotonically with respect to model performance and reaches its maximum value of 1 when the model performs best. With an increase in $F$, MISS is less (more) sensitive to the model's ability to simulate amplitudes (patterns).

In terms of climate model evaluation, the pattern similarity is usually more important than the amplitude, because without pattern similarity, the accuracy of amplitude simulation is often less meaningful. Thus, one can set $F$ to be a value greater than 1 in Eq. (10) for general model evaluation purpose. In this case, MISS/MIEI is more sensitive to the change in the pattern similarity than the amplitude. Considering that MIEI has a geometric meaning when $F$ is 2, which represents the length of a line segment (referring to the line segment CG in Figure 3 in Xu et al., 2017). Thus, 2 appears to be a reasonable value of $F$ for general model evaluation purpose. Users can also change $F$ based on the application. For example, one may

use a smaller $F$, say $F=0.5$, to give more weight to the amplitude if one wants to evaluate model ability to simulate the long-term trend of the multiple variables, e.g. the surface air temperature and specific humidity. In this case, one may have more concern about the values of the trends than their spatial patterns.

## 2.2 Centered statistics

As well as uncentered statistics, centered statistics are also important when the anomaly field is the main concern of model evaluation. For the centered mode, centered RMSL (cRMSL), centered VSC (cVSC), and centered RMSVD (cRMSVD) with area weighting are defined to evaluate the model performance in terms of anomaly fields. The centered statistics are the same as the uncentered statistics, except that the original field is replaced by the anomaly field. These statistics are written as follows:

$$cL_A = \sqrt{\sum_{i=1}^{M}\sum_{j=1}^{N} w_j(a_{ij} - \bar{a}_i)^2}, cL_O = \sqrt{\sum_{i=1}^{M}\sum_{j=1}^{N} w_j(o_{ij} - \bar{o}_i)^2} \tag{11}$$

$$cR_v = \frac{\sum_{i=1}^{M}\sum_{j=1}^{N} w_j(a_{ij} - \bar{a}_i)(o_{ij} - \bar{o}_i)}{cL_A \cdot cL_O} \tag{12}$$

$$cRMSVD = \sqrt{\sum_{i=1}^{M}\sum_{j=1}^{N} w_j[(a_{ij} - \bar{a}_i) - (o_{ij} - \bar{o}_i)]^2} \tag{13}$$

One can use the vector mean error (VME) to additionally measure the difference between two mean vector fields since the mean difference was removed from the centered statistics mentioned above. The VME can also be written as the root-mean-square error of two mean fields:

$$VME = \sqrt{\sum_{i=1}^{M}(\bar{a}_i - \bar{o}_i)^2}, \bar{a}_i = \sum_{j=1}^{N} w_j a_{ij}, \bar{o}_i = \sum_{j=1}^{N} w_j o_{ij} \tag{14}$$

As the uncentered statistics (Eqs. 1–3) can be transformed into centered statistics (Eqs. 11–13), by replacing the original field with the anomaly field, cRMSL, cVSC, and cRMSVD also satisfy the cosine law:

$$cRMSVD^2 = cL_A^2 + cL_O^2 - 2 \cdot cL_A \cdot cL_O \cdot cR_v \tag{15}$$

Furthermore, with the aid of Eq. (3), we can decompose the square of RMSVD as follows:

$$RMSVD^2 = \sum_{i=1}^{M}\sum_{j=1}^{N} w_j\left\{\left[(a_{ij} - \bar{a}_i) - (o_{ij} - \bar{o}_i)\right] + (\bar{a}_i - \bar{o}_i)\right\}^2$$

$$= \sum_{i=1}^{M}\sum_{j=1}^{N} w_j[(a_{ij} - \bar{a}_i) - (o_{ij} - \bar{o}_i)]^2 + \sum_{i=1}^{M}\sum_{j=1}^{N} w_j(\bar{a}_i - \bar{o}_i)^2 + 2 \cdot \sum_{i=1}^{M}\left\{(\bar{a}_i - \bar{o}_i)\sum_{j=1}^{N} w_j\left[(a_{ij} - \bar{a}_i) - (o_{ij} - \bar{o}_i)\right]\right\} \tag{16}$$

With the aid of Eqs. (13)–(14), RMSVD, cRMSVD, and VME satisfy the Pythagorean Theorem:

$$RMSVD^2 = cRMSVD^2 + VME^2 = VME^2 + cL_A^2 + cL_O^2 - 2 \cdot cL_A \cdot cL_O \cdot cR_v \tag{17}$$

Clearly, these statistics for vector variable evaluation satisfy the cosine law and Pythagorean Theorem. Similarly, such relationships are also valid for scalar variables (Taylor, 2001; Xu and Han, 2019).

Similar to rms_std (Eq. 6), the standard deviation of SD (SD_std) is also defined to describe the dispersion of SD over all variables:

$$\sigma_{SD} = \sqrt{\frac{1}{M}\sum_{m=1}^{M}\left(cL_{Am}^* - \frac{1}{M}\sum_{m=1}^{M}cL_{Am}^*\right)^2} \tag{18}$$

where, $cL_{Am}^*$ is the same as $L_{Am}^*$, except that it is the ratio of SDs.

## 3 MVIE with a combination of multiple scalar and vector fields

The MVIE method proposed by Xu et al. (2017) considers only multiple scalar fields. Under some circumstances, one may
want to simultaneously evaluate both scalar and vector fields. Here, the MVIE method is improved to meet this need. Assume there are $M$ individual variables to be evaluated, which are either scalar or vector fields (the upper left part of Fig. 1). Variable $d_m$ is the dimension of the $m$th variable, where $d_m$ is equal to 1 for a scalar field (e.g., temperature) while $d_m$ is 2 for a two-dimensional vector field (e.g., a vector wind), and so on. Hereafter, the vector field for an individual variable (e.g., a 850-hPa wind field) is termed the individual vector field to separate it from vector fields grouped from multiple fields.
Following the idea of MVIE, these variables are normalized with respective rms values of the reference. Note that an individual scalar field is normalized by dividing by its rms value. An individual vector field is normalized as a whole by dividing by its RMSL. These normalized scalar and/or vector fields can be grouped into a multivariable field with the dimension $D \times N$, where $D$ is the sum of $d_m$. The multivariable field derived from the model can be evaluated against that derived from observation by using the various performance metrics in the uncentered or centered mode (Fig. 1).
The uncentered mode focuses on the whole original field, while the centered mode separately evaluates the anomaly field and the mean field. Each mode of statistics consists of three levels of statistics: statistics for individual variables (yellow boxes), multivariable integrated statistics (green boxes), and an index summarizing the overall model performance (orange boxes). The definitions of centered and uncentered statistics are the same as those defined by Xu et al. (2017), except that the area weighting is considered here. To calculate the statistics for individual variables (e.g., root-mean-square difference
(RMSD), centered RMSD (cRMSD) or uncentered CORR (uCORR)), we can also use the formulas of multivariable integrated statistics (Eqs. 1–3, 11–14) by setting $M$ equal to $d_m$, which is summarized in the right-hand part of the boxes in Fig. 1.

Note that the mean error (ME) is additionally computed for the centered statistics, as the centered statistics exclude mean error. For a scalar variable, ME is calculated with Eq. (14) by setting $M$ to 1, but it is signed. Because VME is a function of
175 the difference in the vector magnitude and direction, we provide two additional statistical metrics — the mean error of vector magnitude (MEVM) and the mean error of vector direction (MEVD) — to separate the magnitude error from the directional error. MEVM (MEVD) is the mean of the magnitude error (direction difference) between the modelled vector and the observed vector for all grids evaluated. Note that the MEVD is only valid for 2D vector fields. The direction difference ranges from −180 to 180, and the positive (negative) value represents a counter-clockwise (clockwise) directional error of
180 the model mean vector.

To summarize the overall model skill score in terms of the simulation of multiple variables, the uncentered MISS (uMISS) and centered MISS (cMISS) are provided for the uncentered and centered modes, respectively. uMISS is calculated with Eqs.

(9) and (10) using the original fields. cMISS can also be calculated with Eqs. (9) and (10), but replacing the rms and VSC by SD and cVSC, respectively. With the support of these statistics, the improved MVIE method can provide a more comprehensive and precise evaluation of model performance. All statistics defined in this paper together with their acronyms are summarized in the table of A1 in the Appendix.

## 4 The Multivariable Integrated Evaluation Tool

### 4.1 Brief overview

The Multivariable Integrated Evaluation Tool (MVIETool) consists of two main scripts and some function scripts. All these scripts are written in NCL, which can be easily used in Linux and Macintosh operating systems. The two main scripts are `Calculate_MVIE.ncl` and `Plot_MVIE.ncl`. The execution of the MVIETool can be simplified to two runs, which work independently but in sequence (Fig. 2). Users can modify arguments written in a module at the beginning of the main scripts according to the application. The script assumes that the model data and observation data are saved in Network Common Data Form (NetCDF) format. The `Calculate_MVIE.ncl` script calculates the statistical metrics defined in this paper. The output of this script is saved in a NetCDF file, which is used as the input to `Plot_MVIE.ncl` for plotting the VFE diagram and the metrics table.

### 4.2 Preparing the input data

The MVIETool requires two groups of datasets as inputs — the model data and observations — saved in NetCDF format. Each model or observational data file includes all the variables to be evaluated. If the variables are saved separately as CMIP data, one can easily merge these variables into one data file by using third-party software: e.g., the Climate Data Operator (CDO) or NetCDF operators (NCO). The main script also assumes that all model and observation data files are stored in the same directory. Therefore, users need to move or link various data into the same directory. Variables stored in the data file need to be on the same grid. Examples are given in the User Guide of the MVIETool package to show how to regrid data on regular or irregular grid into the same regular grid with NCL, CDO, and Python3, respectively. In terms of vector variables, each component of the vector variable should be stored independently. If users want to consider area weighting in the statistics, the variables should be saved with the dimension names and the coordinate information (e.g., time, latitude, and longitude), because the coordinate information is needed for the calculation of area weighting. Currently, the tool can only deal with area weighting for regular grids and area weighting is calculated by the formula as:

$$w_j = sin(lat_j + d_{lat}) - sin(lat_j - d_{lat}) \tag{19}$$

where $lat_j$ is the latitude in $jth$ grid and $d_{lat}$ is the difference in latitude between two adjacent zonal grids. The tool can only identify the time and geographical coordinates of regular grid: i.e., time, latitude, longitude, and level.

Figure 3 illustrates an example that consists of ten models (M1–M10) to be evaluated and two sets of reanalysis (REA1, REA2) data as reference in the same directory. Each data file includes eight variables: Q600, SLP, SST, T850, u850, v850, u200, and v200. Among these, u850 (u200) and v850 (v200) are the zonal and meridional components of vector winds in 850-hPa (200-hPa), respectively. The MVIETool allows treating u850 (u200) and v850 (v200) as an individual vector field rather than two scalar fields. To declare a vector field, users can simply put the components of a vector in parenthesis separated by comma, e.g., (u850, v850) and (u200, v200) in the argument `Varname` of the tool (Table 1). Thus, the evaluation actually includes six individual variables. One can also save various surface variables (e.g. SST) and multi-level variables (e.g., air temperature) in the same file. The MVIETool can only evaluate part of the multi-level variables specified by user.

## 4.3 Usage and workflow of the MVIETool

Once datasets have been prepared, one can use the MVIETool to evaluate model performance. Users should set some arguments at the beginning of `Calculate_MVIE.ncl` based on the application. The arguments are summarized in Table 1 and discussed in this section. The rightmost column of Table 1 gives an example of argument setting. Arguments 1–5 define the data file information as in the example of Fig. 3. Note that, in the argument `Varname` in Table 1, the vector variable is identified by enclosing its components in parentheses: e.g., (u200, v200). Notably, if users want to add area weighting to the statistics, the data should have the latitude coordinate, by setting arguments 9–11. Some arguments are mandatory, e.g., argument 1–5 in Table 1, while some arguments are optional. We provide a default value for most of the arguments and the default value will be used if users do not specify the argument, except for `Range_time`, `Coords_geo`, `Range_geo` and `VarLev`. These four arguments must be set, when input variables have corresponding coordinates.

After reading the data, the reference data is calculated with observation and/or reanalysis data. As shown in Fig. 4, if users provide only one observational dataset, it is directly used as the reference data. Otherwise, the mean of multiple observational datasets is used as the reference and each observational piece of data will also be evaluated against the ensemble mean to measure the observational uncertainty. Considering that some variables may contain missing values and some may not, to make the evaluation comparable between different models, a common mask for all models and the reference data is generated to deal with the datasets as the default option. In addition, the tool can also unify the missing points for each model-reference pair separately by modifying the argument `ComMask_On`. Further, whether to unify missing points across all variables of one model can also be chosen with the help of the argument `Unify_VarMiss`.

The script can calculate the statistics either for a single variable or for multiple variables (Fig. 4). The left blue dotted box in Fig. 4 shows the calculation process for single variable evaluation (SVE). The centered and uncentered modes calculate the centered and uncentered statistics, respectively. Note that the calculated statistical metrics rely on the type of input variable. If it is a scalar field, uCORR (CORR), rms (SD), and RMSD (cRMSD) are calculated in the uncentered (centered) mode. With regard to a vector field, VSC (cVSC), RMSL (cRMSL), and RMSVD (cRMSVD) are calculated in the

uncentered (centered) mode. In addition, two skill scores defined by Taylor (2001), $S_1$ and $S_2$, are computed in both the centered and uncentered modes for a scalar variable. Similarly, $S_{v1}$ and $S_{v2}$ are calculated for a vector variable (Xu et al., 2016). After the calculation, these statistics are saved in a file that can be used to generate a Taylor diagram or VFE diagram.

In terms of the MVIE (green dotted box of Fig. 4), the statistics of individual variables (i.e., scalar variables and vector variables) are calculated first. After the evaluation for each individual variable, all variables are normalized by the respective rms values of the reference and are grouped into a multivariable integrated field for the calculation of multivariable statistics. To consider the relative importance of various variables, weights can be added to each variable after normalization through the argument `Wgt_var` (Table 1). In the centered mode, either VME or MEVM is computed for a vector variable and the multivariable field based on the argument `Cal_VME` (Table 1). When MEVM is chosen, if the individual 2D vector variable exists, the MEVD is also calculated for it. Finally, both the centered and uncentered MISS are calculated. If more than one observational dataset is available, the statistics between each observation and the reference are calculated to take the observational uncertainty into account.

After the calculation, the statistics calculated above are written to a new NetCDF file specified by the `MVIE_filename` argument. Meanwhile, the ranges of these statistics can be printed on the screen if the `Print_stats_r` is set to `True`, helping users to set the color levels in the metrics table (Figs. 5, 6).

Similarly, users can modify parameters in `Plot_MVIE.ncl` to control the display of a figure or table. Users can also modify the attribute parameters for the VFE diagram and the metrics table. A detailed explanation and default values can be found in the `Plot_MVIE.ncl` script. Users can choose to create the VFE diagram, the metrics table, or both. Interpretations of plots are discussed in detail with examples in Section 5. In addition, all arguments in the MVIETool and their descriptions are summarized in `readme.namelist` for users' reference. More detailed explanations of the arguments can be found in the User Guide of MVIETool.

## 5 Application of the tool

To illustrate the application of the tool, monthly mean datasets of ten CMIP5 models (Table A2 in the Appendix) derived from the first ensemble run of historical experiments during the period from 1961 to 2000 are used. The variables used include climatological mean 600-hPa specific humidity (Q600), sea level pressure (SLP), sea surface temperature (SST), 850-hPa temperature (T850), 850-hPa 2D vector wind (uv850), and 200-hPa 2D vector wind (uv200) in spring (March–April–May), summer (June–July–August), autumn (September–October–November), and winter (December–January–February). We assessed the model's ability to simulate these variables in the Northern Hemisphere. The mean of two sets of reanalysis data is used as a reference: the Japan Meteorological Agency and the Central Research Institute of Electric Power Industry Reanalysis-55 (JRA55) and the National Centers for Environmental Prediction/National Center for Atmospheric Research Reanalysis Project (NNRP). All datasets are regridded to a common resolution of 2.5° × 2.5° using a bilinear

interpolation method before the evaluation. A common mask of missing value is used for all model and reanalysis datasets in each season.

## 5.1 Metrics table

The metrics table can show various model performance metrics in terms of individual variables and multivariable integrated field, as well as the overall model skill scores in either centered or uncentered mode. Figure 5 shows the metrics table of

various statistical metrics, which evaluates six climatological mean fields — SLP, SST, Q600, T850, uv850, and uv200 — with centered statistical metrics. The filled color of each grid cell represents the value of statistical metric. Lighter colors indicate the model statistics are closer to observation and vice versa. The corresponding color bars can be shown below the metric table such as Fig. 6. Different types of statistics are separated from each other by a thick black line. To facilitate the comparison of the metrics from different variables, in the centered mode, the SD (cRMSL), cRMSD (cRMSVD), and ME

(VME) of the models are normalized by dividing by the corresponding SD (cRMSL) of the reference. In the uncentered mode, rms (RMSL) and RMSD (RMSVD) are normalized using rms (RMSL).

The metrics table of the centered statistics decomposes the original field into mean and anomaly fields for evaluation. The anomaly fields are further evaluated from the perspective of pattern similarity, variance, and overall difference between the model and observation. The metrics table can clearly explain how much of the overall error comes from the mean error (ME,

VME), the amplitude error of the anomaly field (SD, cRMSL), or the error in pattern similarity of the anomaly field (CORR, cVSC). For example, the ME and cRMSD of M1 in simulating SLP are 0.106 and 0.563, respectively, indicating that the overall error is caused mainly by the error in the anomaly field (Fig. 5). The cRMSD can be further attributed to the poor amplitude (1.275) and pattern similarity (0.906), which can be shown more clearly in a Taylor diagram or VFE diagram. Similarly, one can also decompose model errors into mean error (VME) and overall error of the anomaly field (cRMSVD) in

terms of the simulation of multiple variables.

To summarize and rank the overall performance of a model in simulating multiple fields, the MISSs in both centered and uncentered modes are provided in the metrics table and are expected to provide a more accurate evaluation compared with MIEI. Figure 5 shows that M2 ranks eighth out of ten models when referring to the values of the centered MIEI (cMIEI), while it ranks fifth based on the cMISS. The main reason is that cMIEI is sensitive to the error in SDs, particularly for an SD

greater than 1. For example, the SD of SLP in M2 is 1.4 and it contributes about 0.027 to the first term on the right-hand side of Eq. (7a). The cVSC of M2 is 0.954, which contributes about 0.092 to the second term on the right-hand side of Eq. (7a). Referring to the definition of $L^{*}_{Am}$ (Eq. 8), 1.4 is equivalent to its reciprocal, 0.714, in the sense of model performance. However, the SD of 0.714 contributes only about 0.014 to the first term on the right-hand side of Eq. (9). Thus, the MISS is equally sensitive to the model's abilities to simulate pattern similarity and amplitude. In addition, cMISS (uMISS) also

allows us to adjust the relative importance of SD (rms) values and cVSC (VSC) based on the application (Eq. 10).

The MVIETool can be used in a very flexible way by modifying the arguments in the main scripts. A comparison of Fig. 5 with Fig. 6 helps to explain the flexibility. For example, users can choose the statistics to be displayed. In Fig. 6, only a few statistical metrics are displayed in comparison with Fig. 5. Unlike Fig. 5, Fig. 6 divides each grid cell into four triangles, representing model performance in each of the four seasons. Currently, the grid cell can be divided into two or four triangles. If no value of metrics is displayed in the grid cell, colored bars and a box legend are provided for reference. Moreover, considering that the relative number of models compared to the variables and statistical metrics may vary with the application, the MVIETool allows users to transpose the metrics table into a portrait or landscape orientation. For example, the model labels (statistical metrics and variables) can be arranged on the top or the left of the metrics table. More detailed technical introduction for plotting is provided in the User guide.

## 5.2 VFE diagram

The VFE diagram is used to measure the model's ability to simulate the original (anomaly) vector or multiple fields in terms of three statistics: RMSL (cRMSL), VSC (cVSC), and RMSVD (cRMSVD). Figure 7 is the VFE diagram for a vector variable generated by the MVIETool. It assesses the climatological mean 850-hPa vector winds of the Northern Hemisphere in autumn derived from ten CMIP5 models (M1–M10) during the period 1961–2000. Since the anomaly field of 850-hPa vector wind is considered, cRMSL, cVSC, and cRMSVD are shown in the diagram. The construction of the VFE diagram is based on the geometric relationship (Eq. 15) between the three statistics. Thus, in this diagram, the azimuthal position gives VSC (cVSC), the radial distance from the origin indicates RMSL (cRMSL), and the distance between the model and the reference points provides RMSVD (cRMSVD). Similar to the metrics table, RMSL (cRMSL) and RMSVD (cRMSVD) were normalized by the RMSL (cRMSL) of the reference to facilitate an intercomparison between different variables. The VFE diagram can clearly show how much of the overall difference between model and observation is caused by poor pattern similarity and how much is due to the difference in the field amplitude (Xu et al., 2017).

Besides, a red horizontal line is shown in Fig.7 centered at the 'REF' point on X-axis, the length of which can represent the observational uncertainty. Here, we use the area-weighted mean of standard deviations ($M_{SD}$) derived from multiple observations as the estimation of the observational uncertainty:

$$M_{SD} = \frac{\sum_{i=1}^{M}\sum_{j=1}^{N} w_j \cdot SD_{ij}^*}{M \cdot N}, SD_{ij}^* = \frac{SD_{ij}^{obs}}{SD_{ij}^t} \tag{20}$$

where $j$ ($i$) represents the grid (variable) index and $w_j$ is the area weighting. $SD_{ij}^{obs}$ is the standard deviation of multiple observations, which is calculated using the climatologies of REA1 and REA2 (Fig. 7). Clearly, more observational data are desirable to derive a statistically meaningful standard deviation. Here, we only aim to illustrate how to show observational uncertainty in the VFE diagram. $SD_{ij}^t$ represents the inter-annual standard deviation of the reference, which is derived from the 40-year time series in autumn from 1961 to 2000. $M_{SD}$ is illustrated with the red line in Fig. 7 and it summarizes the mean dispersion of multiple observations in all grids for $M$ variables, which can roughly represent the overall uncertainty of observations.

Fig. 7a is the same as Fig. 7b except that Fig. 7a applies area weighting to the statistics, but Fig. 7b does not. Note that for some models (e.g., M2, M4, and M5), there is a relatively large difference between the statistics without and with area weighting, including $M_{SD}$. We recommend taking area weighting into account in model evaluation of a spatial field, especially for regions covering a broad span of latitudes. Furthermore, the tool can also generate a VFE diagram with SD (rms), CORR (uCORR), and cRMSD (RMSD), and in this situation it is the same as the Taylor diagram (Taylor, 2001; Xu et al., 2016).

## 6 Summary

In this paper, we have improved the MVIE method and developed the MVIETool to support the evaluation of model performance using this method. The improved MVIE method can evaluate overall model performance in simulating multiple scalar variables and vector variables. In addition, we consider area weighting in the definition of statistics, which is important for the evaluation of spatial fields on a longitude and latitude mesh grid. Based on MIEI, we further define a more accurate multivariable integrated skill score, termed MISS, to evaluate and rank the overall model performance in simulating multiple variables. Similar to MIEI, MISS also takes both amplitude and the pattern similarity into account, but it is a normalized index with the maximum value of 1 representing a perfect model. MISS is also flexible and able to adjust the relative importance between the pattern similarity and amplitude.

A Multivariable Integrated Evaluation Tool, MVIETool (version 1.0), was developed in NCL code to facilitate the evaluation and intercomparison of model performance. The tool provides two modes of statistics, the uncentered mode and the centered mode, for different requirements of evaluation. The uncentered statistics, such as RMSL, VSC, and RMSVD, evaluate the model performance in terms of the original field. In contrast, the centered statistics evaluate model performance in simulating anomaly fields. In practice, some variables are dependent on each other to a certain extent, such as the 850-hPa and 700-hPa temperatures, and thus contain redundant information. To adjust the relative importance of various variables, we also take into consideration variable weighting in the statistics of the multivariable fields.

The MVIETool primarily consists of two main scripts with one for statistical metrics calculation and the other one for plotting. The tool is programmed to handle network Common Data Form (NetCDF) data as input with a fixed format. Users can control the evaluation by setting arguments at the beginning of the main script. The statistics are shown in the VFE diagram and/or the metrics table, which provide a valuable visual overall evaluation of model performance. We demonstrated the utility of the MVIETool through three examples of ten CMIP5 models in Section 5. The improved MVIE method, together with the MVIETool, are expected to assist researchers to efficiently evaluate model performance in terms of multiple fields.

To make the evaluation methods available to more users, we also develop the MVIETool with Python3. Currently, the MVIETool 1.0 only provides some basic function to calculate statistics and generate figures for MVIE. We will continue to develop the tool to support more comprehensive evaluation. For example, the area weighting is only valid for the regular

grid in MVIETool 1.0. In terms of irregular grids, the area weighting can be derived from an additional data file that contains the grid area of each grid. To address observation uncertainty, the tool compares each individual observation against the average of multiple observations and the spread across various observations is taken as a measure of observational uncertainty. Another approach is to calculate the standard deviation of multiple observations as uncertainty estimation at present, which is also very basic. It warrants further investigation to develop a more sophisticated method that can estimate

the impacts of observational uncertainty on model evaluation. In addition, no significance test is available yet for difference between two vectors fields as well as the multivariable statistics, which also warrants for development in the future.

Furthermore, the Earth System Model Evaluation Tool (ESMValTool; Eyring et al., 2016; Weigel et al., 2020) is a systemic and efficient tool for model evaluation, which has been widely used in related studies (e.g., Valdes et al., 2017; Righi et al., 2020; Waliser et al., 2020). It has many distinct advantages, such as providing the well-documented analysis and

no need for preprocessing of evaluated datasets, compared with our tool. In the follow-up work, we would not only devote to making advance in the function of MVIETool, but also intend to collaborate with the ESMValTool to include our package into it. In this way, users can benefit from the MVIETool with more convenience.

*Code and data availability.* The MVIETool 1.0 (coded with NCL or Python3) is written in open source scripts and is uploaded as a supplement as a frozen version of the MVIETool 1.0. The codes and relevant data to generate figures in this

paper are also provided in the supplement. Additionally, these are also available at https://github.com/Mengzhuo-Zhang/MVIETool from the GitHub repository. Here, we will update the codes with minor improvements and to fix bugs in future.

**Appendix:**

 **Table A1. Table of statistics in the improved multivariable integrated method with their acronyms and descriptions.**

| Acronyms | Description |
| --- | --- |
| *Statistics for individual variables* | |
| SD | Standard deviation |
| CORR | Correlation coefficient |
| cRMSD | Centered root-mean-square difference |
| rms | Root-mean-square |
| uCORR | Uncentered correlation coefficient |
| RMSD | Root-mean-square difference |
| ME | Mean error |
| *Statistics for multivariable integrated field* | |
| cRMSL | Centered root-mean-square length (Eq. 11) |
| cVSC | Centered vector similarity coefficient (Eq. 12) |
| cRMSVD | Centered root-mean-square vector difference (Eq. 13) |
| SD_std | Standard deviation of SD values (Eq. 18) |
| RMSL | Root-mean-square length (Eq. 1) |
| VSC | Vector similarity coefficient (Eq. 2) |
| RMSVD | Root-mean-square vector difference (Eq. 3) |
| rms_std | Standard deviation of rms values (Eq. 6) |
| VME | Vector mean error (Eq. 14) |
| MEVM | Mean error of vector magnitude |
| MEVD | Mean error of vector direction |
| *Index for summarizing overall performance* | |
| MIEI | Multivariable integrated evaluation index (Eq. 7a) |
| cMIEI | Centered multivariable integrated evaluation index |
| MISS | Multivariable integrated skill score |
| cMISS | Centered multivariable integrated skill score |
| uMISS | Uncentered multivariable integrated skill score (Eqs. 8–10) |
| *Observational uncertainty* | |
| $M_{SD}$ | Mean of standard deviation derived from multiple observations (Eq. 20) |

**Table A2. Model names, institution and horizontal resolution for 10 CMIP5 models (M1–M10) used in the paper.**

| | Model | Institution | Horizontal resolution |
|---|---|---|---|
| *M1* | BNU-ESM | College of Global Change and Earth System Science, Beijing Normal University (China) | 2.81° × 2.81° |
| *M2* | CCSM4 | NCAR (National Center for Atmospheric Research) Boulder (USA) | 1.25°× 0.94° |
| *M3* | CNRM-CM5 | Centre National de Recherches Meteorologiques / Centre Europeen de Recherche et Formation Avancees en Calcul Scientifique (France) | 1.41° × 1.41° |
| *M4* | BCC-CSM1-1 | Beijing Climate Center, China Meteorological Administration (China) | 2.81° × 2.81° |
| *M5* | FGOALS-g2 | LASG, Institute of Atmospheric Physics, Chinese Academy of Sciences; and CESS, Tsinghua University (China) | 2.81° × 3.05° |
| *M6* | GFDL-ESM2M | Geophysical Fluid Dynamics Laboratory (USA) | 2.5° × 2.0° |
| *M7* | GISS-E2-H | NASA Goddard Institute for Space Studies (USA) | 2.5° × 2.0° |
| *M8* | MIROC4h | Atmosphere and Ocean Research Institute (The University of Tokyo), National Institute for Environmental Studies, and Japan Agency for Marine-Earth Science and Technology (Japan) | 0. 56° × 0.56° |
| *M9* | MIROC-ESM-CHEM | Atmosphere and Ocean Research Institute (The University of Tokyo), National Institute for Environmental Studies, and Japan Agency for Marine-Earth Science and Technology (Japan) | 2.81° × 2.79° |
| *M10* | inmcm4 | Institute for Numerical Mathematics (Russia) | 2.0° × 1.5° |

*Author contributions.* MZ and ZX devised the evaluation method and wrote the paper. MZ coded the MVIETool 1.0. All the
authors discussed the results and commented on the paper.

*Competing interests.* The authors declare that they have no conflict of interest.

*Acknowledgments.* We thank the climate modelling groups involved in CMIP project for producing and making available
their model outputs. NCEP Reanalysis 2 data was provided by the NOAA/OAR/ESRL PSD, Boulder, Colorado, USA, from
their website at http://www.esrl.noaa.gov/psd/. The JRA55 Reanalysis data was provided from Japanese 55-year reanalysis
projects carried out by the Japan Meteorological Agency (JMA). The study was supported jointly by the National Key
Research and Development Program of China (2018YFA0606004, 2017YFA0603803) and the National Science Foundation
of China (41675105, 41775075, 42075152). This work was also supported by the Jiangsu Collaborative Innovation Center
for Climate Change.

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

**Figure 1. General idea and performance metrics of the improved MVIE.** The left-hand column illustrates the general idea of the MVIE. Two modes of statistics are provided for evaluation: uncentered statistics (middle column) and centered statistics (right-hand column). In each mode, the statistics are sorted into three grades: statistics for individual variables, statistics for the multiple field, and a summary index. The right-hand side of the statistics box is its formula.

**Figure 2. The structure of the MVIETool.** The primary input of the workflow is model data and observation (gray) with fixed formats. Inside the blue dotted boxes are two independent runs for calculation and plotting, respectively. Two main scripts (yellow) need to be invoked in sequence. The outputs (green) are the NetCDF file storing performance metrics, the VFE diagram, and the metrics table.

```
promote:data zhangmengzhuo$ pwd
/Users/zhangmengzhuo/data
promote:data zhangmengzhuo$ ls
example.M1.nc    example.M4.nc    example.M8.nc
example.M10.nc   example.M5.nc    example.M9.nc
example.M2.nc    example.M6.nc    example.REA1.nc
example.M3.nc    example.M7.nc    example.REA2.nc
```

| example.M1.nc | example.M1.nc | Local File |
|---|---|---|
| lat | lat | 1D |
| lon | lon | 1D |
| Q600 | Q600 | Geo2D |
| SLP | SLP | Geo2D |
| SST | SST | Geo2D |
| T850 | T850 | Geo2D |
| u200 | u200 | Geo2D |
| u850 | u850 | Geo2D |
| v200 | v200 | Geo2D |
| v850 | v850 | Geo2D |

**Figure 3. Example of data preparation for the MVIETool.** There are eight variables stored in each data file: Q600, SLP, SST, T850, u200, u850, v200, v850. Variables u850 (u200) and v850 (v200) compose an individual 2D vector variable — uv850 (uv200), and the other four variables are regarded as scalar variables. REA1 and REA2 are two sets of reanalysis data used in the evaluation.

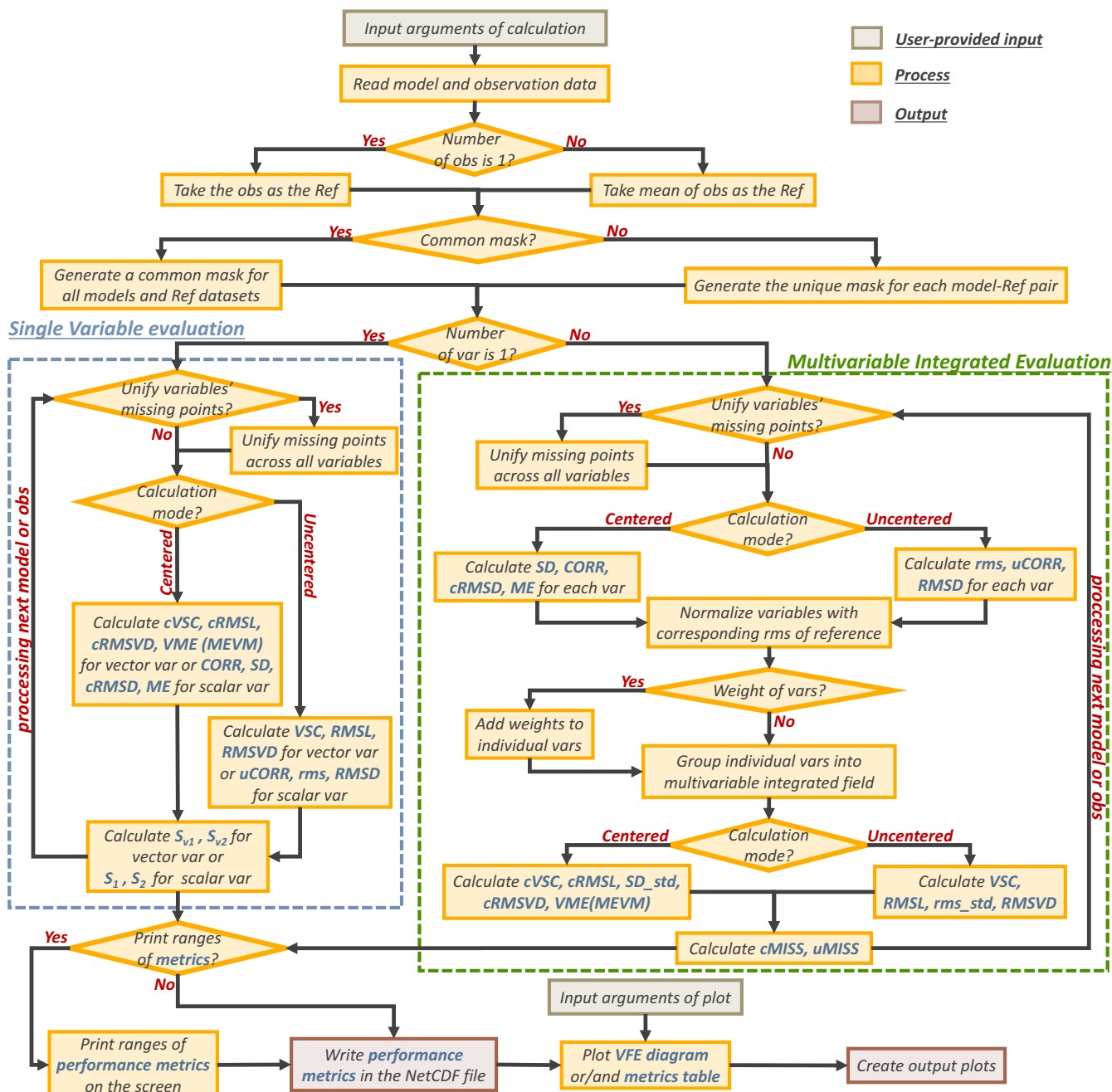

**Figure 4. Flow chart of the MVIETool procedure.** The blue and green dotted rectangles outline the procedures of single variable evaluation and multivariable integrated evaluation, respectively.

| METRICS | | M1 | M2 | M3 | M4 | M5 | M6 | M7 | M8 | M9 | M10 | REA1 | REA2 |
|---|---|---|---|---|---|---|---|---|---|---|---|---|---|
| ME | SLP | 0.106 | 0.068 | 0.025 | 0.041 | 0.395 | 0.711 | -0.397 | 0.014 | 0.364 | -0.059 | 0.004 | -0.004 |
| | SST | 0.034 | 0.043 | 0.008 | -0.010 | -0.094 | 0.013 | 0.074 | 0.093 | 0.059 | -0.011 | -0.002 | 0.002 |
| | Q600 | 0.371 | 0.296 | -0.080 | 0.059 | 0.296 | 0.038 | -0.301 | 0.018 | 0.019 | 0.321 | -0.061 | 0.061 |
| | T850 | -0.026 | 0.026 | -0.022 | -0.077 | -0.139 | -0.151 | -0.037 | 0.114 | 0.005 | -0.037 | 0.009 | -0.009 |
| | uv850 | 0.072 | 0.046 | 0.038 | 0.025 | 0.067 | 0.093 | 0.039 | 0.102 | 0.152 | 0.112 | 0.005 | 0.005 |
| | uv200 | 0.097 | 0.055 | 0.013 | 0.175 | 0.010 | 0.025 | 0.124 | 0.146 | 0.030 | 0.076 | 0.019 | 0.019 |
| VME | | 0.125 | 0.095 | 0.035 | 0.091 | 0.118 | 0.089 | 0.111 | 0.110 | 0.096 | 0.121 | 0.020 | 0.020 |
| cRMSD | SLP | 0.563 | 0.758 | 0.414 | 0.499 | 0.598 | 0.451 | 0.406 | 0.546 | 0.595 | 0.486 | 0.072 | 0.072 |
| | SST | 0.137 | 0.114 | 0.126 | 0.147 | 0.215 | 0.134 | 0.151 | 0.136 | 0.163 | 0.216 | 0.063 | 0.063 |
| | Q600 | 0.468 | 0.445 | 0.258 | 0.316 | 0.410 | 0.387 | 0.297 | 0.352 | 0.395 | 0.626 | 0.100 | 0.100 |
| | T850 | 0.097 | 0.103 | 0.092 | 0.135 | 0.126 | 0.123 | 0.132 | 0.102 | 0.152 | 0.146 | 0.037 | 0.037 |
| | uv850 | 0.483 | 0.454 | 0.311 | 0.437 | 0.476 | 0.414 | 0.432 | 0.396 | 0.655 | 0.517 | 0.119 | 0.119 |
| | uv200 | 0.274 | 0.256 | 0.256 | 0.317 | 0.328 | 0.269 | 0.405 | 0.300 | 0.347 | 0.299 | 0.050 | 0.050 |
| cRMSVD | | 0.352 | 0.331 | 0.242 | 0.326 | 0.360 | 0.310 | 0.342 | 0.303 | 0.452 | 0.400 | 0.086 | 0.086 |
| SD_std | | 0.129 | 0.167 | 0.029 | 0.106 | 0.092 | 0.069 | 0.056 | 0.065 | 0.152 | 0.156 | 0.017 | 0.017 |
| cMIEI | | 0.365 | 0.373 | 0.247 | 0.337 | 0.364 | 0.313 | 0.350 | 0.308 | 0.455 | 0.427 | 0.088 | 0.087 |
| SD | SLP | 1.275 | 1.400 | 1.010 | 1.171 | 0.964 | 1.182 | 0.999 | 1.106 | 1.210 | 1.196 | 1.023 | 0.981 |
| | SST | 1.048 | 1.030 | 0.970 | 1.024 | 1.136 | 0.977 | 0.979 | 0.963 | 0.907 | 0.944 | 0.975 | 1.028 |
| | Q600 | 1.292 | 1.276 | 0.982 | 0.960 | 1.223 | 1.126 | 0.870 | 1.139 | 1.019 | 1.352 | 1.017 | 0.993 |
| | T850 | 0.998 | 0.946 | 0.989 | 1.072 | 1.035 | 1.052 | 1.010 | 0.970 | 0.890 | 0.901 | 1.012 | 0.990 |
| | uv850 | 1.245 | 1.206 | 0.954 | 1.130 | 1.022 | 1.102 | 0.956 | 1.039 | 1.296 | 1.099 | 1.000 | 1.014 |
| | uv200 | 1.005 | 0.977 | 0.920 | 0.856 | 0.977 | 1.013 | 1.051 | 1.056 | 0.980 | 0.990 | 0.985 | 1.017 |
| cRMSL | | 1.127 | 1.094 | 0.958 | 1.036 | 1.050 | 1.059 | 0.984 | 1.028 | 1.085 | 1.043 | 0.997 | 1.010 |
| CORR | SLP | 0.906 | 0.852 | 0.915 | 0.906 | 0.815 | 0.928 | 0.917 | 0.870 | 0.872 | 0.917 | 0.998 | 0.998 |
| | SST | 0.992 | 0.994 | 0.992 | 0.990 | 0.988 | 0.991 | 0.989 | 0.991 | 0.990 | 0.977 | 0.998 | 0.998 |
| | Q600 | 0.948 | 0.952 | 0.966 | 0.949 | 0.952 | 0.941 | 0.959 | 0.954 | 0.924 | 0.901 | 0.995 | 0.995 |
| | T850 | 0.995 | 0.996 | 0.996 | 0.994 | 0.993 | 0.994 | 0.991 | 0.995 | 0.994 | 0.994 | 0.999 | 0.999 |
| | uv850 | 0.930 | 0.932 | 0.951 | 0.923 | 0.890 | 0.927 | 0.904 | 0.925 | 0.868 | 0.883 | 0.993 | 0.993 |
| | uv200 | 0.963 | 0.967 | 0.968 | 0.953 | 0.945 | 0.964 | 0.923 | 0.959 | 0.939 | 0.955 | 0.999 | 0.999 |
| cVSC | | 0.952 | 0.954 | 0.970 | 0.949 | 0.940 | 0.956 | 0.941 | 0.956 | 0.909 | 0.924 | 0.996 | 0.996 |
| cMISS | | 0.960 | 0.960 | 0.980 | 0.963 | 0.957 | 0.968 | 0.959 | 0.969 | 0.934 | 0.943 | 0.997 | 0.998 |
| uMISS | | 0.979 | 0.982 | 0.991 | 0.983 | 0.978 | 0.984 | 0.978 | 0.985 | 0.967 | 0.972 | 0.999 | 0.999 |

**Figure 5. Metrics table of centered statistics.** The table evaluates the performance of ten CMIP5 models in simulating climatological mean (1961–2000) 600-hPa specific humidity (Q600), sea level pressure (SLP), sea surface temperature (SST), 850-hPa temperature (T850), 850-hPa winds (uv850) and 200-hPa winds (uv200) in the Northern Hemisphere in autumn (September–October–November). ME (VME) is the error of the mean scalar or vector (multivariable) fields. cRMSD (cRMSVD) is the overall difference in scalar or vector (multivariable) anomaly fields between model and observation. CORR (cVSC) and SD (cRMSL) are the pattern similarity and amplitude of the anomaly fields for the individual variable (multivariable field). SD_std is the standard deviation of SD values. cMISS (uMISS) is the multivariable integrated skill score of the anomaly (original) fields, which is calculated with SD (rms) values and cVSC (VSC). cMIEI is the centered multivariable integrated index. The factor $F$ in cMISS and uMISS is 2. The colors represent the values of statistical metrics. The lighter colors indicate better model performance.

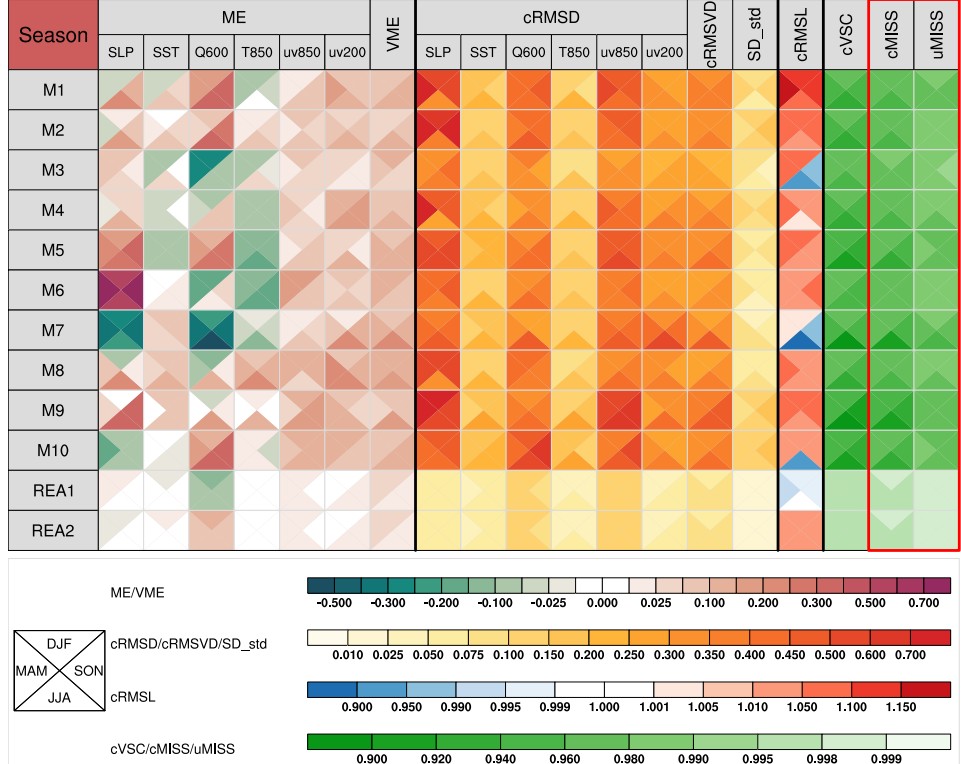

**Figure 6. Metrics table of centered statistics.** These models are used to simulate climatological means of 600-hPa specific humidity (Q600), sea level pressure (SLP), sea surface temperature (SST), 850-hPa temperature (T850), 850-hPa winds (uv850) and 200-hPa winds (uv200), of the Northern Hemisphere in four seasons, spring (March–April–May), summer (June–July–August), autumn (September–October–November), and winter (December–January–February). Each square is divided into four triangles, representing model performance in different seasons, as shown in the bottom-left legend. The colored bars for different statistical metrics are shown below the table.

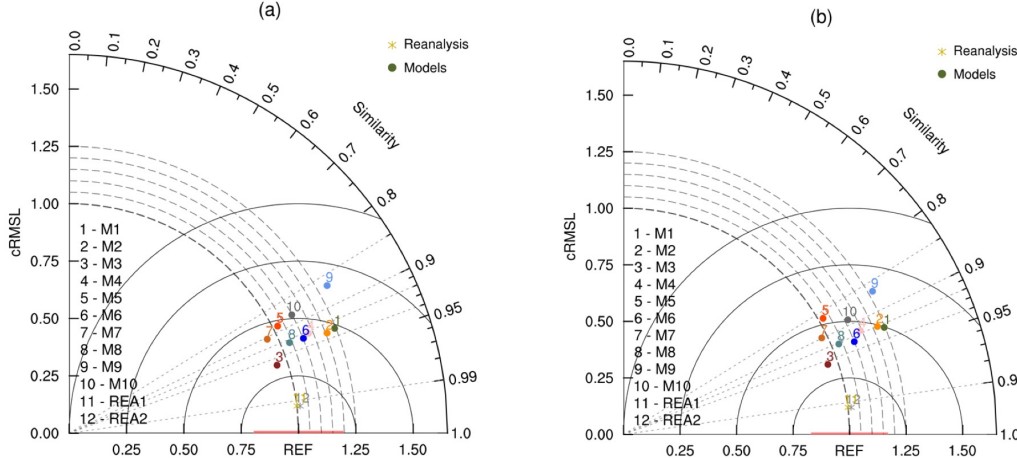

**Figure 7. VFE diagram.** This describes the climatological mean autumn (September–October–November) 850-hPa wind fields derived from the different CMIP5 models (M1–M10) in the Northern Hemisphere. The azimuthal position gives cVSC, the radial distance from the origin indicates cRMSL, and the distance between the model and the reference points provides cRMSVD. The cRSML and the cRMSVD are normalized by the cRMSL derived from the mean of two reanalysis datasets (REA1, REA2). The area weighting is considered in the statistical metrics in (a) but not in (b). The observational uncertainty is indicated by the red horizontal line centered at the REF point. Here, we use $M_{SD}$ value derived from REA1 and REA2 to estimate observational uncertainty.

**Table 1. Input arguments to `Calculate_MVIE.ncl` in the MVIETool.** An argument with [*] is optional. The rightmost column provides examples of these arguments.

| | Argument | Description | Example |
|---|---|---|---|
| 1 | **Varname** | Names of independent variables stored in data file. For individual vector variable, names of its components need to be enclosed in parentheses separated by comma. | `"Q600, SLP, SST, T850, (u200, v200), (u850, v850)"` |
| 2 | **Model_filenames** | Names of model data files for evaluation. | `"example."+(/"M1","M2","M3", "M4","M5","M6","M7","M8","M9 ","M10"/)+".nc"` |
| 3 | **Obs_filenames** | Names of observation and/or reanalysis data files for evaluation. | `"example."+(/"REA1","REA2"/) +".nc"` |
| 4 | **Inputdatadir** | Model and observation data input directory. | `"/Users/zhangmengzhuo/data/"` |
| 5 | **Var_Coords** | Whether to read data by the range of coordinates. | `True` |
| 6 | [*]**isCoords_time** | Whether evaluated variables have time dimension and coordinate, under **Var_Coords** is `True`. | `True` |
| 7 | [*]**Coords_time** | Name of time dimension for evaluated variables, under **isCoords_time** is `True`. | `"time"` |
| 8 | [*]**Range_time** | Range of time in time coordinate of evaluated variables, under **isCoords_time** is `True`. | `(/"19710101","19991201"/)` |
| 9 | [*]**isCoords_geo** | Whether evaluated variables have latitude and/or longitude dimensions and coordinates, under **Var_Coords** is `True`. | `True` |
| 10 | [*]**Coords_geo** | Names of latitude and/or longitude dimensions for evaluated variables, under **isCoords_geo** is `True`. | `(/"lat","lon"/)` |
| 11 | [*]**Range_geo** | Ranges of latitude and/or longitude in their coordinates of evaluated variables, under **isCoords_geo** is `True`. It has the format as: `(/"lat|0:45", "lon|0:180"/)`. | `(/"lat|0:45","lon|0:180"/)` |
| 12 | [*]**hasLevel** | Whether evaluated variables have level dimension. If having, set it to the level dimension name; otherwise set `False`. The level coordinate is required to read data at specific level. | `False` |
| 13 | **\*VarLev** | Specify the level for each evaluated variable. If a variable does not have the level dimension, users can provide an arbitrary value in the corresponding position to match the variable in **Varname**. | – |
| 14 | [*]**Isarea_wgt** | If no area weighting in statistics, set `False`; otherwise set to the name of the latitude dimension in **Coords_geo**. | `"lat"` |

| 15 | *Type_stats | Data type of calculated statistics. | "float" |
|---|---|---|---|
| 16 | *Stats_mode | Calculate uncentered / centered statistics, and it is 1 or 0, representing the uncentered or the centered mode. | 0 |
| 17 | *Wgt_var | Add weights of variables. If not adding, set to 1; otherwise set 1D numeric array, that should match variables in **Varname**. | (/1,1,2,3,1,1,2,2/) |
| 18 | *ComMask_On | Create a common mask for all model data and reference (True), or unify the missing points between each model-reference pair (False). | True |
| 19 | *Unif_VarMiss | Whether to unify missing points across all variables of one model. | True |
| 20 | *Cal_VME | Calculate VME/MEVM in centered mode. Set it to 1 (0), representing calculating VME (MEVM). | 1 |
| 21 | *MISS_F | Parameter $F$ in the calculation of MISS. | 2 |
| 22 | *Print_stats_r | Whether to print range of performance metrics on the screen. | True |
| 23 | *MVIE_filename | Output NetCDF file name. | "example.centered.nc" |