# Peer review of "An improved multivariable integrated evaluation method and NCL code for multimodel intercomparison (MVIETool version 1.0)"

_Geoscientific Model Development, 2020_

## Short Comment (SC1) · 21 Dec 2020

Dear authors,

in my role as Executive editor of GMD, I would like to bring to your attention our Editorial version 1.2:

https://www.geosci-model-dev.net/12/2215/2019/

This highlights some requirements of papers published in GMD, which is also available on the GMD website in the 'Manuscript Types' section: http://www.geoscientific-model-development.net/submission/manuscript_types.html

[Figure]

In particular, please note that for your paper, the following requirement has not been met in the Discussions paper:

- Code must be published on a persistent public archive with a unique identifier for the exact model version described in the paper or uploaded to the supplement, unless this is impossible for reasons beyond the control of authors. All papers must include a section, at the end of the paper, entitled "Code availability". Here, either instructions for obtaining the code, or the reasons why the code is not available should be clearly stated. It is preferred for the code to be uploaded as a supplement or to be made available at a data repository with an associated DOI (digital object identifier) for the exact model version described in the paper. Alternatively, for established models, there may be an existing means of accessing the code through a particular system. In this case, there must exist a means of permanently accessing the precise model version described in the paper. In some cases, authors may prefer to put models on their own website, or to act as a point of contact for obtaining the code. Given the impermanence of websites and email addresses, this is not encouraged, and authors should consider improving the availability with a more permanent arrangement. Making code available through personal websites or via email contact to the authors is not sufficient. After the paper is accepted the model archive should be updated to include a link to the GMD paper.

As GitHub is not a persistent archive, please provide a persistent release for the exact source code version used for the publication in this paper. As explained in https://www.geoscientific-model-development.net/about/manuscript_types.html the preferred reference to this release is through the use of a DOI which then can be cited in the paper. For projects in GitHub a DOI for a released code version can easily be created using Zenodo, see https://guides.github.com/activities/citable-code/ for details. Finally note, that according to our new Editorial (v1.2) all data and analysis / plotting scripts should be made available.

Yours, Astrid Kerkweg

---

## Author Comment (AC1) · 24 Dec 2020

Dear Editor,

Thank you for your comments. To meet the requirement of GMD, we uploaded the code, data, and scripts used to generate the figures on Zenodo with a doi 10.5281/zenodo.4387038. We can also upload the code and sample data as a supplement (less than 20M) during the manuscript revision.

Although we carefully checked out the code and applied it to model evaluations. The code may still contain bugs. If it is acceptable, we would like to keep the GitHub link

in the code availability section after uploading the code and data as a supplement. In doing so, we can update the code in GitHub in the future if we or users find new bugs.

Yours,

Zhongfeng Xu on behalf of all Co-Authors

Institute of Atmospheric Physics,

Chinese Academy of Sciences

---

## Referee Comment (RC1) · Anonymous Referee #1 · 27 Dec 2020

I have carefully read the paper about the development of a new evaluation method for multiple fields and multimodels by Zhang et al, and, despite the fact that I think that the paper is mostly well written (language, structure, and so on), I can not recommend its acceptance in its present form.

In terms of scientific significance, I found the paper poor, since it basically uses a very simple technique (weighted average) to re-use techniques that have been published in the past.

In terms of scientific quality, I find that many references are missing, the authors do not consider techniques that have been common in climatology in the last twenty years,

presenting them as advanced. I will develop this point later in detail.

Regarding scientific reproducibility, the authors use some models as example (M1 to M10) and two reanalyses (REA1 and REA2) without mentioning the models, the re-analyses used, the periods, the experiments

For me, the main concern is related to the novelty (or lack of) of the paper. As the authors properly recognize in their section 2.1, the majority of the new methodology involved in the diagram has already been published in two papers such as Xu et al. (2016) and Xu et al., (2016). Thus, as far as I can see, and as written by the authors in the abstract, the new developments in this paper refer to:

1. The use of area-weighting by means of the use of a weighted average

2. The extension of their code to a potential combination of scalar and vector fields. Which, as explained by the authors in Figure 1, involves the change in the dimensions of the input matrix to their evaluation method.

Regarding point 1 above, the authors make what I find a very misleading statement in line 44-45 of their paper, I quote "most previous model performance metrics did not consider spatial weight". This is clearly not true. The paper by Taylor (2001) which gave rise to the idea of the Taylor diagram and which was cited by the authors, already mentions the possibility to use weighted statistics (see page 7183, lines after Eq (1) in that paper). Moreover, Boer and Lambert (2001) thoroughly cover this idea and explicitly used weights w_k in their formulation. The use of the square root of the cosine to account for the varying size of grid points in the estimation of EOFs goes back as far as North et al., (1982), at least, and is commonly used (see the description of function eofcov() in NCL, the programming language used by the authors in their implementations). Additional examples in the use of weights in the evaluation of climate models to account for different grid points can be found elsewhere such as Eq. (1) in Gleckler et al. (2008) or seminal papers in the field such as Reichler and Kim (2008). Studies can be found explicitly devoted to the analysis of the role that smoothing plays

in the verification statistics (Mason and Knutti, 2011; Räisänen and Ylhäisi, 2011). The fact that meridional grid size can be misleading in the evaluation of climate models is well known since at least Benestad (2005). Thus, I think that the authors can not state that the consideration of different weight factors for different grid points to account for their different sizes as written in their paper is novel. And, by itself, the use of a weighted mean instead of a simple mean, does not seem very advanced, either. So, I can not recommend the acceptance of the paper on the basis of this being an advance in science, since this has been constantly carried out in papers during the last twenty years.

Second, the combination of multiple fields (or components of vector fields) as presented in point 2 above can also be a problem, from my point of view. As I see it, the algorithm lumps in the same indices (points in the diagram) information from different variables or components of different vector fields. Even though it might be practical to have a single model-evaluation index (point in their diagram), the fact that different variables are mixed might be obscuring important diagnostics. For instance, vector variables can show differences in the orientation of the simulated vector fields or their relative variances. I'd suggest the authors to discuss this issue by presenting (for instance) the way that two similar synthetic vector datasets behave if their error statistics are similar but they differ in the way the error statistics are distributed in the zonal and meridional directions, for instance. This would highlight the way these statistics are reflected in the diagram designed by authors. I guess that if the same amount of error is distributed in the zonal/meridional directions in two synthetic models, the authors are going to get the very same points in their diagram, but the source of the error is very different.

Finally, the authors highlight in substantial parts of their manuscript that they provide an implementation of their methodology using NCL. This is apparently an important part of their contribution, since it is stated so in the abstract, section 4 and Table 1. However, NCL has been kept in maintenance mode by NCAR

https://www.ncl.ucar.edu/open_letter_to_ncl_users.shtml since September 2019 and this is not mentioned in the manuscript.

I understand that the implementation of the technique provides a tool "ready to go" for climate scientists, but I doubt this is enough for a highly cited journal such as GMD. However, may be I am wrong and the editor thinks otherwise. For me, the difference between a rejection or a major revision is just a matter of how much the editor thinks a "ready to use" tool is a valid contribution. I am not used to the editorial policies of GMD, so that this finally ends in his/her hands.

Benestad, R. E. (2005), On latitudinal profiles of zonal means, Geophys. Res. Lett., 32, L19713, doi:10.1029/2005GL023652.

Boer, G., Lambert, S. Second-order space-time climate difference statistics. Climate Dynamics 17, 213–218 (2001). https://doi.org/10.1007/PL00013735

P. J. Gleckler, K. E. Taylor, and C. Doutriaux (2008) Performance metrics for climate models, JOURNAL OF GEOPHYSICAL RESEARCH, VOL. 113, D06104, doi:10.1029/2007JD008972

Masson, D., & Knutti, R. (2011). Spatial-Scale Dependence of Climate Model Performance in the CMIP3 Ensemble, Journal of Climate, 24(11), 2680-2692.

North, G. R., Bell, T. L., Cahalan, R. F., & Moeng, F. J. (1982). Sampling Errors in the Estimation of Empirical Orthogonal Functions, Monthly Weather Review, 110(7), 699-706.

Räisänen, J., & Ylhäisi, J. S. (2011). How Much Should Climate Model Output Be Smoothed in Space?, Journal of Climate, 24(3), 867-880.

Reichler, T., & Kim, J. (2008). How Well Do Coupled Models Simulate Today's Climate?, Bulletin of the American Meteorological Society, 89(3), 303-312.

Xu, Z. and Hou, Z. and Han, Y. and Guo, W. (2016) A diagram for evaluating multiple

aspects of model performance in simulating vector fields, Geoscientific Model Development, 94365–4380 10.5194/gmd-9-4365-2016

Xu, Z. and Han, Y. and Fu, C. (2017) Multivariable integrated evaluation of model performance with the vector field evaluation diagram, Geoscientific Model Development, 10:3805–3820, doi: 10.5194/gmd-10-3805-2017

---

## Author Comment (AC2) · 6 Jan 2021

Thank you very much for your careful reading and comments. Our point-by-point responses are as follows:

=================================================================

Reviewer #1

For me, the main concern is related to the novelty (or lack of) of the paper. As the authors properly recognize in their section 2.1, the majority of the new methodology involved in the diagram has already been published in two papers such as Xu et al. (2016) and Xu et al., (2017). Thus, as far as I can see, and as written by the authors in the abstract, the new developments in this paper refer to:1. The use of area-weighting by means of the use of a weighted average 2. The extension of their code to a potential combination of scalar and vector fields. Which, as explained by the authors in Figure 1, involves the change in the dimensions of the input matrix to their evaluation method. Regarding point 1 above, the authors make what I find a very misleading statement in line 44-45 of their paper, I quote "most previous model performance metrics did not consider spatial weight". This is clearly not true. The paper by Taylor (2001) which gave rise to the idea of the Taylor diagram and which was cited by the authors, already mentions the possibility to use weighted statistics (see page 7183, lines after Eq (1) in that paper). Moreover, Boer and Lambert (2001) thoroughly cover this idea and explicitly used weights w_k in their formulation. The use of the square root of the cosine to account for the varying size of grid points in the estimation of EOFs goes back as far as North et al., (1982), at least, and is commonly used (see the description of function eofcov() in NCL, the programming language used by the authors in their implementations). Additional examples in the use of weights in the evaluation of climate models to account for different grid points can be found elsewhere such as Eq. (1) in Gleckler et al. (2008) or seminal papers in the field such as Reichler and Kim (2008). Studies can be found explicitly devoted to the analysis of the role that smoothing plays in the verification statistics (Mason and Knutti, 2011; Räisänen and Ylhäisi, 2011). The fact that meridional grid size can be misleading in the evaluation of climate models is well known since at least Benestad (2005). Thus, I think that the authors cannot state that the consideration of different weight factors for different grid points to account for their different sizes as written in their paper is novel. And, by itself, the use of a weighted mean instead of a simple mean, does not seem very advanced, either. So, I cannot recommend the acceptance of the paper on the basis of this being an advance in science, since this has been constantly carried out in papers during the last twenty years.

**Response:**

Many thanks for introducing the references regarding the statistics that considered area-weighting. We will make a further discussion on this issue in the revised manuscript afterward. We agree with the reviewer that the sentence "*most previous model performance metrics did not consider spatial weight*" is inappropriate. Some statistical metrics did consider area-weighting and some were not. We will revise the sentence as "*The statistical metrics employed in Xu et al., (2016; 2017) did not consider spatial weight*". The detailed responses to the reviewer's comment are as follows:

As the reviewer pointed out that area-weighting was considered in previous statistical metrics, e.g., correlation coefficient, standard deviation (e.g., Watterson, 1996; Boer and Lambert, 2001; Masson and Knutti, 2011). These statistical metrics were designed to evaluate *scalar fields*. However, the statistical metrics employed in our previous papers (Xu et al., 2016; 2017), e.g., vector similarity coefficient (VSC), root-mean-square vector length (RMSL), and root-mean-square vector difference (RMSVD), which were devised to evaluate vector fields. To our knowledge, VSC and RMSL were *firstly defined* in our paper and the *area weight was not yet considered* (Xu et al., 2016). With these statistical metrics, we constructed a vector field evaluation (VFE) diagram, which can be regarded as a generalized Taylor diagram (Xu et al., 2016). The VFE diagram can be used to evaluate model performance in simulating vector fields or multiple variable fields with centered or uncentered statistics. In contrast, the Taylor diagram is a special case of the VFE diagram when the VFE diagram is applied to a scalar field with centered statistics. In the GMD manuscript, we redefine the VSC, RMSL, and RMSVD by taking area-weighting into account. More importantly, the three statistical metrics still satisfy the cosine law after considering area weight, which underpins the construction of the VFE or Taylor diagram. Thus, we can take area-weighting into account in the metrics that measuring vector statistics.

Previous studies, e.g., Taylor (2001), Boer and Lambert (2001), Gleckler et al. (2008), did mention or explicitly introduce area weight in the statistical metrics. However, these metrics are generally used to measure scalar fields rather than the vector fields. Thus, *the consideration of area-weighting in the definition of vector field statistics* is one of the novelty of this study relative to previous studies including

our previous studies (Taylor, 2001; Boer and Lambert, 2001;Gleckler et al., 2008; Xu et al., 2016; 2017).

Regarding the comment that "meridional grid size can be misleading in the evaluation of climate models is well known since at least Benestad (2005)", our responses are as follows: It is important to consider the effective sample size in the comparison of zonal mean between different latitudes (Benestad et al., 2011; Forland et al., 2011; Parding et al., 2019). However, in terms of model evaluation, we usually focus on the inter-comparison between various models rather than between different latitudes. All models are evaluated over the same domain and with the same horizontal resolution. Thus, the impact of meridional grid size on model evaluation should be less important after taking area-weighting into account.

The study of spatial smooth in climate model evaluation (Masson and Kuntti, 2011) is very interesting. Similar studies can also be carried out with the statistical metrics defined in our manuscript to investigate the impact of spatial smooth on the overall model performance in simulating multiple fields. This will be discussed in the section of discussion and conclusions in the revised manuscript.

Second, the combination of multiple fields (or components of vector fields) as presented in point 2 above can also be a problem, from my point of view. As I see it, the algorithm lumps in the same indices (points in the diagram) information from different variables or components of different vector fields. Even though it might be practical to have a single model-evaluation index (point in their diagram), the fact that different variables are mixed might be obscuring important diagnostics. For instance, vector variables can show differences in the orientation of the simulated vector fields or their relative variances. I'd suggest the authors to discuss this issue by presenting (for instance) the way that two similar synthetic vector datasets behave if their error statistics are similar but they differ in the way the error statistics are distributed in the zonal and meridional directions, for instance. This would highlight the way these statistics are reflected in the diagram designed by authors. I guess that if the same amount of error is distributed in the zonal/meridional directions in two synthetic models, the authors are going to get the very same points in their diagram, but the source of the error is very different.

**Response:**

Thanks for the reviewer's insightful comment. We agree with the comment that 'Even though it might be practical to have a single model-evaluation index (point in their diagram), the fact that different variables are mixed might be obscuring important diagnostics'. This issue was discussed in our previous paper (Xu et al., 2017, page 3811, the paragraph about Eq. 21). We also discussed this issue in the section of summary and conclusion in Xu et al. (2017). For example, "*Unavoidably, the higher level of metrics (refer to the vector field evaluation or multivariable integrated evaluation metrics) loses detailed statistical information in contrast to the lower level of metrics (refer to the statistics for individual scalar field). To provide a more comprehensive evaluation of model performance, one can show the VFE diagram together with a table of statistical metrics (Table 1) or other model performance metrics as needed.*"

As the single model-evaluation index, which summarizes multiple statistics of multiple fields, can obscure detailed diagnostics, we included the statistical metrics of the individual scalar and vector variables (e.g., CORR, SD) in addition to the multivariable integrated evaluation index in the metric table (Table 1 in the GMD manuscript and Table 1 in Xu et al., 2017). Thus, the metric table can provide a more comprehensive evaluation of model performance.

On the other hand, the statistics for a scalar variable, e.g., correlation coefficient (CORR), standard deviation (SD), or root mean square difference (RMSD), may also obscure important diagnostics to a certain extent. For example, assuming we have two cases (Fig. R1), both have three time series from Model A, Model B, and observation O. In both cases, Models A and B have the same RMSD, SD, and CORR relative to observation. Thus, Models A and B will overlap with each other in the Taylor diagram. However, the time series in Models A and B show piecewise amplitude difference (Fig. R1(a)) and phase difference (Fig. R1(b)) from each other, respectively. Such errors are not captured by the statistical metrics, either. A similar issue also exists in any other statistical metrics, e.g., the Model Climate Performance Index (MCPI) defined by the average relative error of each variable (Gleckler, et al., 2008) and the Model Performance Index (Reichler et al., 2008). It is impossible to have one index that can measure or capture all errors of a model. Nonetheless, an index that can summarize the overall model performance is still very useful, especially for ranking models (Jury et al., 2014; Sidorenko et al., 2015, 2019; Rackow et al., 2019; Semmler et al., 2020). As shown in the metrics table in the manuscript,

the model with a higher multivariable integrated skill score (MISS) generally shows good performance in simulating individual variables, indicating the rationality of MISS.

Finally, the authors highlight in substantial parts of their manuscript that they provide an implementation of their methodology using NCL. This is apparently an important part of their contribution, since it is stated so in the abstract, section4 and Table 1. However, NCL has been kept in maintenance mode by NCAR https://www.ncl.ucar.edu/open_letter_to_ncl_users.shtml since September 2019 and this is not mentioned in the manuscript. I understand that the implementation of the technique provides a tool "ready to go" for climate scientists, but I doubt this is enough for a highly cited journal such as GMD. However, may be I am wrong and the editor thinks otherwise. For me, the difference between a rejection or a major revision is just a matter of how much the editor think as "ready to use" tool is a valid contribution. I am not used to the editorial policies of GMD, so that this finally ends in his/her hands.

**Response:**

Thanks for the comments. We noticed that NCL has been kept in maintenance mode with no update since 2019. The NCL team still prepares maintenance releases containing critical bug fixes and user-contributed code. Meanwhile, the migration from NCL to Python is still underway. Lots of scientists and studies are familiar with NCL and still using NCL. NCL is still one of the most popular software in the community of climate science. The MVIETool provides users a convenient climate model evaluation tool for NCL users. Moreover, the NCL code and sample data also help readers to understand and test the method and develop their own codes with other computer languages.

On the other hand, we have started developing MVIETool scripts with Python, which is expected to be ready to use within one or two months for climate scientists as well as scientists from other disciplines.

**Reference**
Benestad R. E., Senan R., Balmaseda M., Ferranti L., Orsolini Y. and Melsom A.: Sensitivity of summer 2-m temperature to sea ice conditions, Tellus A: Dynamic Meteorology and Oceanography, 63:2, 324-337, DOI: 10.1111/j.1600-0870.2010.00488.x, 2011.
Boer, G., Lambert, S. Second-order space-time climate difference statistics. ClimateDynamics 17, 213–218 (2001). https://doi.org/10.1007/PL00013735.

Forland E. J., Benestad R., Hanssen-Bauer I., Haugen J. E., and Skaugen T. E.: Temperature and Precipitation Development at Svalbard 1900–2100[J]. Advances in Meteorology, 2011(17).

Gleckler P. J., Taylor K. E., and Doutriaux C., 2008: Performance metrics for climate models, Journal of Geophysical Research Atmospheres, 2008, 113, D06104, doi: 10.1029/2007JD008972.

Jury M. W., Prein A. F., Truhetz H., and Gobiet A., 2014: Evaluation of CMIP5 Models in the Context of Dynamical Downscaling over Europe[J]. Journal of Climate, 2015, 28(14):5575-5582.

Masson. D., Knutti. R., Spatial-Scale Dependence of Climate Model Performance in the CMIP3 Ensemble, Journal of Climate, 24(11), 2680-2692.

Parding K. M., Benestad R., Mezghani A., and Erlandsen H. B.: Statistical Projection of the North Atlantic Storm Tracks, Journal of Applied Meteorology and Climatology 58, 7; 10.1175/JAMC-D-17-0348.1, 2019.

Rackow T., Sein D., Semmler T., Danilov S., Koldunov N. V., Sidorenko D., Wang Q., and Jung T., 2018: Sensitivity of deep ocean biases to horizontal resolution in prototype CMIP6 simulations with AWI-CM1.0, Geosci. Model Dev., 12, 2635–2656, 2019, https://doi.org/10.5194/gmd-12-2635-2019.

Räisänen, J., & Ylhäisi, J. S. (2011). How Much Should Climate Model Output BeSmoothed in Space?, Journal of Climate, 24(3), 867-880.

Reichler, T., and J. Kim, 2008: How well do coupled models simulate today's climate? Bull. Amer. Meteor. Soc., 89, 303–311, doi:10.1175/BAMS-89-3-303.

Semmler T., Danilov S., Gierz P., Goessling H. F., Hegewald J., Hinrichs C., Koldunov. N., Khosravi N., Mu L., Rackow T., Sein D. V., Sidorenko D., Wang Q., and Jung T., 2020: Simulations for CMIP6 With the AWI Climate Model AWI-CM-1-1, Journal of Advances in Modeling Earth Systems, 12, e2019MS002009. https://doi.org/ 10.1029/2019MS002009.

Sidorenko D., Goessling H. F., Koldunov N. V., Scholz P., Danilov S., Barbi D., et al., 2019: Evaluation of FESOM2.0 coupled to ECHAM6.3: Preindustrial and HighResMIP simulations. Journal of Advances in Modeling Earth Systems, 11, 3794–3815. https://doi.org/10.1029/2019MS001696.

Sidorenko D., Rackow T., Jung T., Semmler T., Barbi D., Danilov S., et al., 2015: Towards multi-resolution global climate modeling with ECHAM6–FESOM. Part I: Model formulation and mean climate. Climate Dynamics, 44(3-4), 757–780. https://doi.org/10.1007/s00382-014-2290-6

Taylor, K. E.: Summarizing multiple aspects of model performance in a single diagram, J. Geophys. Res., 106, 7183–7192, 2001.

Watterson, I. G.: NON-DIMENSIONAL MEASURES OF CLIMATE MODEL PERFORMANCE, Int. J. Climatol., 16(4), 379–391, 1996.

Xu, Z., Han, Y., and Fu, C.: Multivariable Integrated Evaluation of Model Performance with the Vector Field Evaluation Diagram, Geosci. Model Dev., 10, 3805–3820, 2017.

Xu, Z., Hou, Z., Han, Y., and Guo, W.: A diagram for evaluating multiple aspects of model performance in simulating vector fields, Geosci. Model Dev., 9, 4365–4380, https://doi.org/10.5194/gmd-9-4365-2016, 2016.

**Figure**

[Figure]

**Figure R1.** Two examples illustrate model errors that are not captured by the commonly used statistical metrics. Each example is composed of three time series from idealized model A (blue upper triangle), model B (green lower triangle), and observation O (orange circle), respectively. Compared to O, Model A and B show different errors, but they have the same RMSD, SD, and CORR.

---

## Referee Comment (RC2) · Anonymous Referee #2 · 20 Jan 2021

The authors describe the extension of a method to assess the performance of climate models in simulating scalar or vector fields based on the concept of the vector field evaluation (VFE) first introduced by Xu et al. (2016). In addition, the authors describe a method summarize a model's ability to simulate multiple variables by introducing the multivariable integrated skill score (MISS).

The manuscript is generally well written and I suggest minor revisions to the manuscript before publication in Geoscientific Model Development addressing the points given below. I do not agree with reviewer #1 that the paper does not provide enough novelty for publication in GMD. In my opinion, extending the widely used Taylor

diagrams to include area weighting and proposing a new integrated measure of a model's performance across variables while giving the user the possibility to adjust the relative importance of RMS and VSC is very welcome. Compiling all metrics into a tool for model evaluation and making it available to other users is worth publishing this description of the tool and the methods used.

I do think, however, that the descriptions of the methods and the tool itself lack some detail. I also have the impression that the MVIETool does not live up to its full potential and could strongly benefit from implementing the routines into the framework of existing model evaluation software. I see this, however, as a potential future pathway and not as a prerequisite for publication. More detail is given in the general comments below.

**General comments**

- Regridding and masking are important processing steps that are not explained in enough detail. For example, it is not clear to me whether all variables from the same source (model or observations) have to be on the same grid (horizontal and vertical). Simply referring to external software such as CDO is not enough and a concrete example should be given (e.g. to reproduce the figures shown). The same is true for masking of missing values. How is this done? For example, is a mask generated for each time step and each dataset? Are all datasets used to create a common mask that is then applied to all datasets or is the mask created separately for each model-observations pair? If masks are generated separately for each model-observations pair, what does this mean for the comparability across different models? It should become clear how data have to be

preprocessed and which implications this might have on comparing across differ-
ent models and/or observational datasets. I recommend adding some discussion
on this issue.

- Considering observational uncertainties in model evaluation is of fundamental
  importance. The approach taken here by using the average of possibly available
  multiple observational datasets as reference data seems very basic. What effect
  does this averaging have on the skill scores? I would expect this kind of averaging
  to reduce the spatial and/or temporal variability of the reference data compared to
  the individual observational datasets and thus have an impact on the skill scores.
  Also, are there ways to include possibly available uncertainty information on a per
  pixel basis (e.g. standard error provided with some ESA CCI satellite datasets)?
  At least a brief discussion on thoughts on this topic should be added.

- Is there a way to visualize observational uncertainty e.g. in the VFE diagrams
  in addition to showing individual observational datasets against the reference
  dataset, e.g. by shading the area representing the uncertainty range in the dia-
  grams?

- A weighting factor $F$ has been introduced but is not discussed. In l. 430, the
  authors state that "The factor F in cMISS and uMISS is 2". What is the reasoning
  for this choice? What is recommended to users wanting to apply the MVIETool?
  Maybe give some examples for specific applications.

- How is the grid cell area calculated that is used as weighting factor? Again, the
  statement (l. 189/190) that "If users want to consider area weighting in the statis-
  tics, the variables should be saved with the coordinate information (e.g., time,
  latitude, and longitude)" does not provide enough detail. How do the coordinates
  have to be defined? Is following the CF standard sufficient? Do the coordinates
  have to follow the CMOR conventions? Can area files ("fx" files in CMIP) be
  provided e.g. for irregular grids or is the analysis limited to regular grids?

- It is also not clear to me how time series of variables are handled. For example, additional information on (possibly) selecting a user specified time range is needed. Are attributes of the time coordinate such as calendar taken into account when calculating time means (e.g. number of days per month)? Is temporal interpolation done if the time resolution of two datasets does not match?

- I was missing an overview (e.g. a table) on which models, experiments, years, time resolution, etc. of the model data and which reanalysis datasets have been used to create the example figures. This makes it impossible to reproduce the examples as an independent check (i.e. downloading the data yourself, applying the preprocessing steps and running the MVIETool) that the software is working as expected.

- I have the impression that the MVIETool would benefit substantially from taking advantage of the infrastructure of existing model evaluation tools such as, for instance, the ESMValTool (Righi et al., Geosci. Model Dev., 2020). Such tools provide the possibility to preprocess all datasets in a consistent way regarding checking of input data, horizontal and vertical regridding, masking, time selection, vertical level selection, etc. I would like to encourage the authors to add some discussion on such a step as a possible outlook to the summary section.

- It becomes increasingly more important to provide traceable and reproducible results. For model evaluation, this usually means providing a provenance record of the input data, used software, configuration, processing steps, etc. Is anything like this planned for the MVIETool? Again, I feel that the MVIETool could strongly benefit from taking advantage of the infrastructure of existing model evaluation tools that are already capable of providing provenance records.

- Are there plans to continue development of MVIETool? I would recommend to add an outlook and thoughts about possible future directions to the summary section.

**Specific comments**

- l. 16, "MIEI" has not been defined yet

- l. 44, "most previous model performance metrics did not consider spatial weight": while this statement is true for the three examples mentioned, this is not the case for many other metrics and therefore needs rephrasing

- l. 65, "... by dividing the corresponding rms value of the observation ...': it is not entirely clear to me what is divided by what, please consider rephrasing; if the original variable is divided by the rms of the observations, are all data on the same spatial and temporal grid? If not, what are the effects of this? What is the effect of averaging possibly available multiple observational datasets (see also general comments)?

- l. 188: "variable" → "variables"

- l. 196/197: What is meant by "put in parenthesis"? Where does this have to be done (source code, namelist, etc.)? Please be more specific.

- l. 215: What is meant by "standardize the missing points"? Does this mean a common mask is created from all missing grid cells/time steps in all datasets (models + reference) across all variables? Please give more details on how masking is done (see also general comments).

- l. 233: "one piece of observational data" → e.g. "one observational dataset"?

- l. 234: "written in a new NetCDF file" → "written to a new NetCDF file"

- l.258/258: What is meant by "better model performance" and "worse model performance"? Better of worse relative compared to what? Please be more specific and if possible more quantitative.

- l. 272: "relative" → "compared"
* * *

---

## Author Comment (AC3) · 29 Jan 2021

Thanks very much for the insightful comments. The comments are very helpful not only for improving this manuscript but also for our future study. Our point-by-point responses are as follows: ==================================================== Reviewer #2 Regridding and masking are important processing steps that are not explained in enough detail. For example, it is not clear to me whether all variables from the same source (model or observations) have to be on the same grid (horizontal and vertical). Simply referring to external software such as CDO is not enough and a concrete

example should be given (e.g. to reproduce the figures shown). The same is true for masking of missing values. How is this done? For example, is a mask generated for each time step and each dataset? Are all datasets used to create a common mask that is then applied to all datasets or is the mask created separately for each model-observations pair? If masks are generated separately for each model-observations pair, what does this mean for the comparability across different models? It should become clear how data have to be preprocessed and which implications this might have on comparing across different models and/or observational datasets. I recommend adding some discussion on this issue. Response: Thanks for the reviewer's comment. We will add more detailed explanations on the regridding and masking of missing values in the revised manuscript. Our responses to these questions are as follows: The MVIETool requires all the variables on the same grid for datasets of all models and observations. We will give an example to illustrate how to regrid data with CDO in an updated pdf file (graphic_guide_MVIETool.pdf). The regridding can be done with one command line of CDO. For example, the following CDO command can interpolate input data (time,lev,lat,lon) to a $1.25° \times 1.25°$ longitude-latitude grid and 14 vertical pressure levels: cdo remapbil, r288x145 –intlevel,100000,925000,85000,70000,60000, 50000,40000,30000,25000,20000,15000,10000,7000,5000 input.nc output.nc In terms of masking, the present MVIETool generates a common mask for each model-observation pair. The mask is the same for all variables in this pair of data. If more than one observational data are available, the tool will generate a common mask for the missing grids first for the observational datasets before the evaluation. In doing so, we can take full advantage of the model output. However, it makes the evaluation less comparable between different models. In the revised manuscript, we will generate a common mask for all models and observation as a default option.

Considering observational uncertainties in model evaluation is of fundamental importance. The approach taken here by using the average of possibly available multiple observational datasets as reference data seems very basic. What effect does this averaging have on the skill scores? I would expect this kind of averaging to reduce the

spatial and/or temporal variability of the reference data compared to the individual observational datasets and thus have an impact on the skill scores. Also, are there ways to include possibly available uncertainty information on a per pixel basis (e.g. standard error provided with some ESA CCI satellite datasets)? At least a brief discussion on thoughts on this topic should be added. Is there a way to visualize observational uncertainty e.g. in the VFE diagrams in addition to showing individual observational datasets against the reference dataset, e.g. by shading the area representing the uncertainty range in the diagrams? Response: Thanks for the reviewer's comments about observation uncertainties in model evaluation. The average of multiple observation datasets may reduce the spatial and/or temporal variability of the reference data compared to the individual observational datasets. This impact can be roughly estimated by the individual observational datasets in the VFE diagram. For example, the cRMSL is slightly greater than 1 for REA1 (point 11) and REA2 (point 12) in Fig. 7 of the manuscript, which indicates that the average of two reanalysis data leads to a slight reduction in spatial variability. However, this reduction is very small and its impacts on the evaluation should be neglectable. If the cRMSDs of individual observational data are clearly greater than 1 for individual observation, one should not use the average of multiple observation datasets as reference. Currently, users can use one of the observational data as reference. This issue will be discussed in the revised manuscript. We are also considering other ways to estimate the observational uncertainty. A preliminary idea is as follows: Assuming we have K observational dataset. Based on K observational data, we can compute the VFE statistics K times and get K points in the VFE diagram for one model. Afterwards, we create a shaded patch using the K points. The area of the patch can represent the impact of observational uncertainty on the evaluation. The approach may only work well when the VFE diagram only show a few models. Otherwise, it would be hard to discriminate one model from others if many shaded patches overlap together. In terms of the observational data that already has an uncertainty estimation (standard deviation in each grid point), we will add a horizontal bar centered at reference point on the x-axis. The length the bar represents the mean standard deviation of the observation, which can roughly represent the mean spread of observational datasets.

A weighting factor F has been introduced but is not discussed. In l. 430, the authors state that "The factor F in cMISS and uMISS is 2". What is the reasoning for this choice? What is recommended to users wanting to apply the MVIETool? Maybe give some examples for specific applications. Response: Thanks for the comment. MISS is defined based on the MIEI (Eq. 7a in the manuscript) proposed by Xu (2017). MISS is equal to MIEI when factor F is 2. MIEI represents the length of line segment CG in Figure 3 in Xu et al., (2017). MIEI (MISS) has a geometric meaning when F is equal to 2. Meanwhile, the MISS is more sensitive to the changes in VSC than RMSL when F is equal to 2. As climate models can often reasonably reproduce the pattern similarity, it is getting harder to improve pattern correlation when the correlation coefficient is greater than 0.8 or higher. Thus, it is usually more desirable to give VSC more weight. F is a flexible factor; user can modify its value based on the applications. We will make more discussion on the factor F in the revised paper.

How is the grid cell area calculated that is used as weighting factor? Again, the statement (l. 189/190) that "If users want to consider area weighting in the statistics, the variables should be saved with the coordinate information (e.g., time, latitude, and longitude)" does not provide enough detail. How do the coordinates have to be defined? Is following the CF standard sufficient? Do the coordinates have to follow the CMOR conventions? Can area files ("fx" files in CMIP) be provided e.g. for irregular grids or is the analysis limited to regular grids? Response: Thanks for the comment about the weighting factor in MVIETool. At present, the tool can only deal with the area weighting for regular grids and the area weighting is calculated by the tool with the equation sin(lat+dlat)- sin(lat-dlat), where dlat is the grid distance in latitude. Hence, variables should first be defined with dimension names and assign the coordinate variables (referring to http://www.ncl.ucar.edu/Document/Language/cv.shtml). The CF standard is sufficient. The requirement for coordinate information will be explained in the revised

paper. In addition to determining the area weight, the coordinate information is also used to select sub-regions for a regional evaluation. One has to regrid all data (model and reanalysis) into a common resolution before the computation of statistical metrics. Thus, one can regrid all data (in a regular or irregular grid) to a regular grid with coordinate information.

It is also not clear to me how time series of variables are handled. For example, additional information on (possibly) selecting a user specified time range is needed. Are attributes of the time coordinate such as calendar taken into account when calculating time means (e.g. number of days per month)? Is temporal interpolation done if the time resolution of two datasets does not match? Response: Thanks for the comment. The MVIETool assumes all input variables on the same grid and with the same coordinate variables, including time, latitude and longitude grid. Otherwise, the processing will break out and report error. In terms of the selection of time range, the tool provides two options: First, one can specify the time range with strings in the format: "YYYYMM", "YYYYYMMDD", or "YYYYMMD-DHH", where YYYY is the year, MM is the month, DD is the day, and HH is the hour, e.g., "198101:199012". In this case, the coordinate variable assigned to the time dimension of input variables should have a "calendar" attribution (referring to http://www.ncl.ucar.edu/Document/Functions/Built-in/cd_calendar.shtml). Second, one can specify the time range with the time steps, e.g., "1:10". We will illustrate how to setting time coordinate in the pdf file (graphic_guide_MVIETool.pdf) as well.

I was missing an overview (e.g. a table) on which models, experiments, years, time resolution, etc. of the model data and which reanalysis datasets have been used to create the example figures. This makes it impossible to reproduce the examples as an independent check (i.e. downloading the data yourself, applying the preprocessing steps and running the MVIETool) that the software is working as expected. Response: Thanks for the comment. The reason we did not give the model name in the manuscript is that the ranks of the models' performance depend on the variables, seasons, and regions evaluated. The model showing good (or poor) performance does not necessarily mean a good (or poor) performance for other variables, seasons and regions. We show the examples to illustrate the methods rather than evaluate specific models. We can present the model name, institution and horizontal resolution of 10 CMIP5 models used in the revised manuscript (attached in Table.R1 in the supplement) if it is necessary. We used the monthly mean datasets derived from the first ensemble run of historical experiments during the period from 1961 to 2000 (L244).

I have the impression that the MVIETool would benefit substantially from taking advantage of the infrastructure of existing model evaluation tools such as, for instance, the ESMValTool (Righi et al., Geosci. Model Dev., 2020). Such tools provide the possibility to preprocess all datasets in a consistent way regarding checking of input data, horizontal and vertical regridding, masking, time selection, vertical level selection, etc. I would like to encourage the authors to add some discussion on such a step as a possible outlook to the summary section. It becomes increasingly more important to provide traceable and reproducible results. For model evaluation, this usually means providing a provenance record of the input data, used software, configuration, processing steps, etc. Is anything like this planned for the MVIETool? Again, I feel that the MVIETool could strongly benefit from taking advantage of the infrastructure of existing model evaluation tools that are already capable of providing provenance records. Are there plans to continue development of MVIETool? I would recommend to add an outlook and thoughts about possible future directions to the summary section. Response: Thanks for the reviewer's constructive comments. We will carefully consider these suggestions in our future work. Currently, the MVIETool only provides some basic functions to calculate the relevant statistics and generate figures. We will try to take advantage of the infrastructure of existing model evaluation tools for further improvement. Our follow-up work intends to devise a significance test method for the difference between two vector fields (and two MISSs). If we can make significant progress, we will incorporate these significant tests into the MVIETool in the future. An outlook of plans and future directions of the MVIETool will be added to the summary section in the revised paper.

L. 16: "MIEI" has not been defined yet Response: The MIEI will be defined in the revised manuscript.

L. 44: "most previous model performance metrics did not consider spatial weight": while this statement is true for the three examples mentioned, this is not the case for many other metrics and therefore needs rephrasing Response: Thanks for the comment. We will revise the sentence as "The statistical metrics employed in Xu et al., (2016; 2017) did not consider spatial weight".

L. 65: "... by dividing the corresponding rms value of the observation ...": it is not entirely clear to me what is divided by what, please consider rephrasing; if the original variable is divided by the rms of the observations, are all data on the same spatial and temporal grid? If not, what are the effects of this? What is the effect of averaging possibly available multiple observational datasets (see also general comments)? Response: Thanks for the comment. The MVIETool requires all input data on the same spatial and temporal grid. We will rephrase the sentence as "We need to normalize each variable derived from the model by dividing the rms value of the corresponding variable derived from observation"

L. 188: "variable" →"variables" Response: We will replace "variable" with "variables" in the revised paper.

L. 196/197: What is meant by "put in parenthesis"? Where does this have to be done (source code, namelist, etc.)? Please be more specific. Response: The MVIETool treats the variables in parenthesis as a vector field, e.g. (ua, va). The variables in the parenthesis represent the different components of a vector field. These variable names are specified in the namelist part of the tool. This will be clarified in the revised manuscript.

L. 215: What is meant by "standardize the missing points"? Does this mean a common mask is created from all missing grid cells/time steps in all datasets (models + reference) across all variables? Please give more details on how masking is done (see

also general comments). Response: Yes, it means generating a common mask for the missing grid. We will reword the sentence and explain more details on the generation of masks in the revised manuscript.

L. 233: "one piece of observational data"→ e.g. "one observational dataset"? Response: We will make the replacement in the revised paper.

L. 234: "written in a new NetCDF file" →"written to a new NetCDF file" Response: We will make the replacement in the revised paper.

L.258/258: What is meant by "better model performance" and "worse model performance"? Better or worse relative compared to what? Please be more specific and if possible more quantitative. Response: Thanks for the comment. "better model performance" and "worse model performance" are determined based on the rank of model performance metrics. The color represents the value of the model performance metrics. This will be clarified in the revised manuscript.

L. 272: "relative" →"compared" Response: We will replace "relative" with "compared" in the revised paper.

Reference Xu, Z., Han, Y., and Fu, C.: Multivariable Integrated Evaluation of Model Performance with the Vector Field Evaluation Diagram, Geosci. Model Dev., 10, 3805–3820, 2017. Xu, Z., Hou, Z., Han, Y., and Guo, W.: A diagram for evaluating multiple aspects of model performance in simulating vector fields, Geosci. Model Dev., 9, 4365–4380, https://doi.org/10.5194/gmd-9-4365-2016, 2016.

Please also note the supplement to this comment:
https://gmd.copernicus.org/preprints/gmd-2020-310/gmd-2020-310-AC3-
supplement.pdf

─────────────────────────

**Supplement:**

**Table**

Table R1. Model names, institution and horizontal resolution for 10 CMIP5 models (M1–M10) used in the paper.

| | Model | Institution | Horizontal resolution |
|---|---|---|---|
| **M1** | BNU-ESM | College of Global Change and Earth System Science, Beijing Normal University(China) | $2.81° \times 2.81°$ |
| **M2** | CCSM4 | NCAR (National Center for Atmospheric Research) Boulder(USA) | $1.25° \times 0.94°$ |
| **M3** | CNRM-CM5 | Centre National de Recherches Meteorologiques / Centre Europeen de Recherche et Formation Avancees en Calcul Scientifique(France) | $1.41° \times 1.41°$ |
| **M4** | BCC-CSM1-1 | Beijing Climate Center, China Meteorological Administration(China) | $2.81° \times 2.81°$ |
| **M5** | FGOALS-g2 | LASG, Institute of Atmospheric Physics, Chinese Academy of Sciences; and CESS, Tsinghua University(China) | $2.81° \times 3.05°$ |
| **M6** | GFDL-ESM2M | Geophysical Fluid Dynamics Laboratory(USA) | $2.5° \times 2.0°$ |
| **M7** | GISS-E2-H | NASA Goddard Institute for Space Studies(USA) | $2.5° \times 2.0°$ |
| **M8** | MIROC4h | Atmosphere and Ocean Research Institute (The University of Tokyo), National Institute for Environmental Studies, and Japan Agency for Marine-Earth Science and Technology(Japan) | $0.56° \times 0.56°$ |
| **M9** | MIROC-ESM-CHEM | Atmosphere and Ocean Research Institute (The University of Tokyo), National Institute for Environmental Studies, and Japan Agency for Marine-Earth Science and Technology(Japan) | $2.81° \times 2.79°$ |
| **M10** | inmcm4 | Institute for Numerical Mathematics(Russia) | $2.0° \times 1.5°$ |

---

## Author Response (AR1)

**Responses to Reviewers**

**Paper number:** gmd-2020-310

**Paper title:** An improved multivariable integrated evaluation method and NCL code for multimodel intercomparison (MVIETool version 1.0)

**Paper authors:** Meng-Zhuo Zhang, Zhongfeng Xu, Ying Han and Weidong Guo

We appreciate the insightful comments and valuable suggestions from the two reviewers, which are very helpful not only for improving this manuscript but also for our future study. Our modifications are highlighted in green in the revised manuscript. Our point-by-point responses are as follows:

**Reviewer #1**

===============================================================

*I have carefully read the paper about the development of a new evaluation method for multiple fields and multimodels by Zhang et al, and, despite the fact that I think that the paper is mostly well written (language, structure, and so on), I can not recommend its acceptance in its present form. In terms of scientific significance, I found the paper poor, since it basically uses a very simple technique (weighted average) to re-use techniques that have been published in the past. In terms of scientific quality, I find that many references are missing, the authors do not consider techniques that have been common in climatology in the last twenty years presenting them as advanced. I will develop this point later in detail. Regarding scientific reproducibility, the authors use some models as example (M1 to M10) and two reanalysis (REA1 and REA2) without mentioning the models, the re-analyses used, the periods, the experiments.*

**Response:**

Thanks for the affirmation for the writing of the manuscript! As to the datasets used in the manuscript, we used CMIP5 datasets derived from the first ensemble run of historical experiments during the period from 1961 to 2000 (Line262 on Page 9). We have presented the model name, institution and horizontal resolution of 10 CMIP5 models in Table A2 in Appendix of the revised paper, or see Table R1 (at the end of response letter) for convenience. Two datasets of reanalysis are the Japan Meteorological Agency and the Central Research Institute of Electric Power Industry Reanalysis-55 (JRA55) and the National Centers for Environmental Prediction/National Center for Atmospheric Research Reanalysis Project (NNRP) (L266–269 on Page 9).

*For me, the main concern is related to the novelty (or lack of) of the paper. As the authors properly recognize in their section 2.1, the majority of the new methodology involved in the diagram has already been published in two papers such as Xu et al. (2016) and Xu et al., (2017). Thus, as far as I can see, and as written by the authors in the abstract, the new developments in this paper refer to:1. The use of area-weighting by means of the use of a weighted average 2. The extension of their code to a potential combination of scalar and vector fields. Which, as explained by the authors in Figure 1, involves the change in the dimensions of the input matrix to their evaluation method. Regarding point 1 above, the*

*authors make what I find a very misleading statement in line 44-45 of their paper, I quote "most previous model performance metrics did not consider spatial weight". This is clearly not true. The paper by Taylor (2001) which gave rise to the idea of the Taylor diagram and which was cited by the authors, already mentions the possibility to use weighted statistics (see page 7183, lines after Eq (1) in that paper). Moreover, Boer and Lambert (2001) thoroughly cover this idea and explicitly used weights w_k in their formulation. The use of the square root of the cosine to account for the varying size of grid points in the estimation of EOFs goes back as far as North et al., (1982), at least, and is commonly used (see the description of function eofcov() in NCL, the programming language used by the authors in their implementations). Additional examples in the use of weights in the evaluation of climate models to account for different grid points can be found elsewhere such as Eq. (1) in Gleckler et al. (2008) or seminal papers in the field such as Reichler and Kim (2008). Studies can be found explicitly devoted to the analysis of the role that smoothing plays in the verification statistics (Mason and Knutti, 2011; Räisänen and Ylhäisi, 2011). The fact that meridional grid size can be misleading in the evaluation of climate models is well known since at least Benestad (2005). Thus, I think that the authors cannot state that the consideration of different weight factors for different grid points to account for their different sizes as written in their paper is novel. And, by itself, the use of a weighted mean instead of a simple mean, does not seem very advanced, either. So, I cannot recommend the acceptance of the paper on the basis of this being an advance in science, since this has been constantly carried out in papers during the last twenty years.*

**Response:**

We agree with the reviewer that the sentence "*most previous model performance metrics did not consider spatial weight*" is inappropriate and have revised the sentence as " the statistical metrics employed in Xu et al., (2016; 2017) did not consider spatial weight" in Lines 44–45 on Page 2. The reviewer pointed out that our manuscript lacks of novelty with regard to the area-weighting because previous studies have already considered area-weighting in statistical metrics (e.g., Watterson, 1996; Boer and Lambert, 2001; Masson and Knutti, 2011). However, these statistical metrics mentioned by the reviewer (e.g., correlation coefficient, standard deviation) are designed to evaluate *scalar fields rather than vector fields*. The statistical metrics in our manuscript are mainly vector field statistics, e.g., vector similarity coefficient (VSC), root-mean-square vector length (RMSL), and root-mean-square vector

difference (RMSVD), which did not yet consider the *area weight* (i.e., Xu et al., 2016; 2017). VSC, RMSL and RMSVD can construct a vector field evaluation (VFE) diagram, which is very useful to evaluate model performance in simulating vector fields or multiple variable fields. In contrast, Taylor diagram is a special case of VFE diagram when the VFE diagram is applied to a scalar field with centered statistics. Hence, taking area-weighting into the definition of VSC, RMSL, and RMSVD is of great importance and makes evaluation more accurate. More importantly, considering area-weight in the definition does not change the relationship between the three statistics and they still satisfy the cosine law, which underpins the construction of the VFE or Taylor diagram. In our point of view, *the consideration of area-weighting in the definition of "vector field statistics"* is one of the novelty of this study relative to previous studies (Taylor, 2001; Boer and Lambert, 2001; Gleckler et al., 2008; Xu et al., 2016; 2017).

Regarding the comment that "*meridional grid size can be misleading in the evaluation of climate models is well known since at least Benestad (2005)*", our response as follows: It is important to consider the effective sample size in the comparison of zonal mean between different latitudes (Benestad et al., 2011; Forland et al., 2011; Parding et al., 2019). In terms of model evaluation, we usually focus on the inter-comparison between different models rather than between different latitudes. All models are evaluated over the same domain and with the same horizontal resolution. Under this circumstance, the impact of meridional grid size on model evaluation should be less important after taking area-weighting into account.

*Second, the combination of multiple fields (or components of vector fields) as presented in point 2 above can also be a problem, from my point of view. As I see it, the algorithm lumps in the same indices (points in the diagram) information from different variables or components of different vector fields. Even though it might be practical to have a single model-evaluation index (point in their diagram), the fact that different variables are mixed might be obscuring important diagnostics. For instance, vector variables can show differences in the orientation of the simulated vector fields or their relative variances. I'd suggest the authors to discuss this issue by presenting (for instance) the way that two similar synthetic vector datasets behave if their error statistics are similar but they differ in the way the error statistics are distributed in the zonal and meridional directions, for*

*instance. This would highlight the way these statistics are reflected in the diagram designed by authors. I guess that if the same amount of error is distributed in the zonal/meridional directions in two synthetic models, the authors are going to get the very same points in their diagram, but the source of the error is very different.*

**Response:**

Thanks for the reviewer's insightful comments. We agree with the comment that "*Even though it might be practical to have a single model-evaluation index (point in their diagram), the fact that different variables are mixed might be obscuring important diagnostics*". This issue was discussed in our previous paper (Xu et al., 2017, page 3811, the paragraph about Eq. 21). We also discussed this issue in the section of summary and conclusion in Xu et al. (2017). For example, "*Unavoidably, the higher level of metrics (refer to the vector field evaluation or multivariable integrated evaluation metrics) loses detailed statistical information in contrast to the lower level of metrics (refer to the statistics for individual scalar field). To provide a more comprehensive evaluation of model performance, one can show the VFE diagram together with a table of statistical metrics (Table 1) or other model performance metrics as needed.*" As the single model-evaluation index summarizing multiple statistics of multiple fields can obscure detailed diagnostics, we included the statistical metrics of the individual scalar and vector variables (e.g., CORR, SD) in addition to the multivariable integrated index in the metric table (Table 1 in the GMD manuscript and Table 1 in Xu et al., 2017). Thus, the metric table can provide a more comprehensive evaluation of model performance.

On the other hand, *any statistics for evaluation may also obscure important diagnostics to a certain extent*, even the statistics for a scalar variable, in that they summarize the error of each value in model data. As illustrated in Fig. R1, models A and B have the same RMSD, SD, and CORR relative to observation, but the piecewise amplitude difference (Fig. R1a) and phases difference (Fig. R1b) between time series of two models cannot be captured by the statistical metrics. It is impossible to have one index that can measure or capture all errors of a model. Nonetheless, an index that can summarize the overall model performance is still very useful, especially for ranking models in many related studies (e.g., Jury et al., 2014; Sidorenko et al., 2015, 2019; Rackow et al., 2019; Semmler et al., 2020). As shown in the metrics table in the manuscript, the model with higher multivariable integrated skill score (MISS) generally shows good performance in simulating individual variables, indicating the rationality of MISS.

*Finally, the authors highlight in substantial parts of their manuscript that they provide an implementation of their methodology using NCL. This is apparently an important part of their contribution, since it is stated so in the abstract, section4 and Table 1. However, NCL has been kept in maintenance mode by NCAR https://www.ncl.ucar.edu/open_letter_to_ncl_users.shtml since September 2019 and this is not mentioned in the manuscript. I understand that the implementation of the technique provides a tool "ready to go" for climate scientists, but I doubt this is enough for a highly cited journal such as GMD. However, may be I am wrong and the editor thinks otherwise. For me, the difference between a rejection or a major revision is just a matter of how much the editor think as "ready to use" tool is a valid contribution. I am not used to the editorial policies of GMD, so that this finally ends in his/her hands.*

**Response:**

Thanks for the comments. One of our goals is to provide a convenient tool to support climate model evaluation. We noticed that NCL has been kept in maintenance mode with no update since 2019 and the NCL team still prepares maintenance releases containing critical bug fixes and user-contributed code. Meanwhile, the migration from NCL to Python is still underway. Lots of scientists and studies are still using NCL, indicating that NCL is still one of the most popular software in the community of climate science. This is why we firstly develop the MVIETool in NCL language. On the other hand, following the advice from the reviewer, we *have already developed MVIETool scripts coded with Python3 language*, which have the same function as NCL version. The MVIETool written in both NCL and Python3 can be easily used by most users in the climate model evaluation community.

**Reference**

Benestad R. E., Senan R., Balmaseda M., Ferranti L., Orsolini Y. and Melsom A.: Sensitivity of summer 2-m temperature to sea ice conditions, Tellus A: Dynamic Meteorology and Oceanography, 63:2, 324-337, DOI: 10.1111/j.1600-0870.2010.00488.x, 2011.

Boer, G., Lambert, S. Second-order space-time climate difference statistics. ClimateDynamics 17, 213–218 (2001). https://doi.org/10.1007/PL00013735.

Forland E. J., Benestad R., Hanssen-Bauer I., Haugen J. E., and Skaugen T. E.: Temperature and Precipitation Development at Svalbard 1900－2100[J]. Advances in Meteorology, 2011(17).

Gleckler P. J., Taylor K. E., and Doutriaux C., 2008: Performance metrics for climate models, Journal of Geophysical Research Atmospheres, 2008, 113, D06104, doi: 10.1029/2007JD008972.

Jury M. W., Prein A. F., Truhetz H., and Gobiet A., 2014: Evaluation of CMIP5 Models in the Context of Dynamical Downscaling over Europe[J]. Journal of Climate, 2015, 28(14):5575-5582.

Masson. D., Knutti. R., Spatial-Scale Dependence of Climate Model Performance in the CMIP3 Ensemble, Journal of Climate, 24(11), 2680-2692.

Parding K. M., Benestad R., Mezghani A., and Erlandsen H. B.: Statistical Projection of the North Atlantic Storm Tracks, Journal of Applied Meteorology and Climatology 58, 7; 10.1175/JAMC-D-17-0348.1, 2019.

Rackow T., Sein D., Semmler T., Danilov S., Koldunov N. V., Sidorenko D., Wang Q., and Jung T., 2018: Sensitivity of deep ocean biases to horizontal resolution in prototype CMIP6 simulations with AWI-CM1.0, Geosci. Model Dev., 12, 2635–2656, 2019, https://doi.org/10.5194/gmd-12-2635-2019.

Semmler T., Danilov S., Gierz P., Goessling H. F., Hegewald J., Hinrichs C., Koldunov. N., Khosravi N., Mu L., Rackow T., Sein D. V., Sidorenko D., Wang Q., and Jung T., 2020: Simulations for CMIP6 With the AWI Climate Model AWI-CM-1-1, Journal of Advances in Modeling Earth Systems, 12, e2019MS002009. https://doi.org/ 10.1029/2019MS002009.

Sidorenko D., Rackow T., Jung T., Semmler T., Barbi D., Danilov S., et al., 2015: Towards multi-resolution global climate modeling with ECHAM6–FESOM. Part I: Model formulation and mean climate. Climate Dynamics, 44(3-4), 757–780. https://doi.org/10.1007/s00382-014-2290-6

Taylor, K. E.: Summarizing multiple aspects of model performance in a single diagram, J. Geophys. Res., 106, 7183–7192, 2001.

Watterson, I. G.: NON-DIMENSIONAL MEASURES OF CLIMATE MODEL PERFORMANCE, Int. J. Climatol., 16(4), 379–391, 1996.

Xu, Z., Han, Y., and Fu, C.: Multivariable Integrated Evaluation of Model Performance with the Vector Field Evaluation Diagram, Geosci. Model Dev., 10, 3805–3820, 2017.

Xu, Z., Hou, Z., Han, Y., and Guo, W.: A diagram for evaluating multiple aspects of model performance in simulating vector fields, Geosci. Model Dev., 9, 4365–4380, https://doi.org/10.5194/gmd-9-4365-2016, 2016.

**Figure**

[Figure]

**Figure R1.** Two examples illustrate model errors that are not captured by the commonly used statistical metrics. Each example is composed of three time series from idealized model A (blue upper triangle), model B (green lower triangle), and observation O (orange circle), respectively. Compared to O, Model A and B show different errors, but they have the same RMSD, SD, and CORR.

**Reviewer #2**

==============================================================

*The authors describe the extension of a method to assess the performance of climate models in simulating scalar or vector fields based on the concept of the vector field evaluation (VFE) first introduced by Xu et al. (2016). In addition, the authors describe a method summarize a model's ability to simulate multiple variables by introducing the multivariable integrated skill score (MISS).*

*The manuscript is generally well written and I suggest minor revisions to the manuscript before publication in Geoscientific Model Development addressing the points given below. I do not agree with reviewer #1 that the paper does not provide enough novelty for publication in GMD. In my opinion, extending the widely used Taylor diagrams to include area weighting and proposing a new integrated measure of a model's performance across variables while giving the user the possibility to adjust the relative importance of RMS and VSC is very welcome. Compiling all metrics into a tool for model evaluation and making it available to other users is worth publishing this description of the tool and the methods used.*

*I do think, however, that the descriptions of the methods and the tool itself lack some detail. I also have the impression that the MVIETool does not live up to its full potential and could strongly benefit from implementing the routines into the frame work of existing model evaluation software. I see this, however, as a potential future pathway and not as a prerequisite for publication. More detail is given in the general comments below.*

**Response:**

Many thanks for the positive comments and support to our study!

*General comments*

*Regridding and masking are important processing steps that are not explained in enough detail. For example, it is not clear to me whether all variables from the same source (model or observations) have to be on the same grid (horizontal and vertical). Simply referring to external software such as CDO is not enough and a concrete example should be given (e.g. to reproduce the figures shown). The same is true for masking of missing values. How is this done? For example, is a mask generated for each time step and each dataset? Are all datasets used to create a common mask that is*

*then applied to all datasets or is the mask created separately for each model-observations pair? If masks are generated separately for each model-observations pair, what does this mean for the comparability across different models? It should become clear how data have to be preprocessed and which implications this might have on comparing across different models and/or observational datasets. I recommend adding some discussion on this issue.*

*L. 215: What is meant by "standardize the missing points"? Does this mean a common mask is created from all missing grid cells/time steps in all datasets (models + reference) across all variables? Please give more details on how masking is done (see also general comments).*

**Response:**

(1) The MVIETool requires the input data to be regridded on the same grid, which is clarified in Lines 198–199 on Page 7, i.e., "Variables stored in the data file need to be on the same grid.". In the User Guide of the MVIETool, we give some examples about how to regrid the input datasets using NCL and Python3 language as well as CDO.

(2) In terms of masking, the previous MVIETool only generated a common mask for each model-observation pair, which takes full advantage of the model output for the evaluation but does hinder the comparability across different models to a certain extent. In the updated MVIETool, a common mask for all datasets can be generated to deal with the missing values. User can select to generate a common mask of missing points for all dataset (default option) or each model-observation pair by setting the argument `ComMask_On`. Please see Lines 229–233 on Page 8 of the revised manuscript. The modification is also attached below:

"Considering that some variables may contain missing values and some may not, to make the evaluation comparable between different models, a common mask for all models and the reference data is generated to deal with the datasets as the default option. In addition, the tool can also unify the missing points for each model-observation pair separately by modifying the argument `ComMask_On`. No matter what kind of masks chosen, the missing points across all variables of one model are the same."

*Considering observational uncertainties in model evaluation is of fundamental importance. The approach taken here by using the average of possibly available multiple observational datasets as reference data seems very basic. What effect does this averaging have on the skill scores? I would expect this kind of averaging to reduce the spatial and/or temporal variability of the reference data compared to the individual observational datasets and thus have an impact on the skill scores. Also, are there ways to include possibly available uncertainty information on a per pixel basis (e.g. standard error provided with some ESA CCI satellite datasets)? At least a brief discussion on thoughts on this topic should be added.*

*Is there a way to visualize observational uncertainty e.g. in the VFE diagrams in addition to showing individual observational datasets against the reference dataset, e.g. by shading the area representing the uncertainty range in the diagrams?*

**Response:**

Thanks for the reviewer's thoughtful comments and suggestions about observation uncertainties in model evaluation. Indeed, the average of multiple observation datasets may reduce the spatial and/or temporal variability of the reference data compared to the individual observational datasets. This impact can be roughly estimated by the points representing the individual observational datasets in the VFE diagram. For example, the cRMSL is slightly greater than 1 for REA1 (point 11) and REA2 (point 12) in Fig. 7 of the manuscript, which indicates that the average of two reanalysis data leads to a slight reduction in spatial variability. However, this reduction is very small and its impacts on the evaluation should be neglectable. If the cRMSD of the individual observational data are clearly greater than 1, one should not use the average of multiple observation datasets as the reference. In the revised manuscript, we add a red horizontal bar centered at reference point on the X-axis of the VFE diagram. The length of red bar represents the mean standard deviation across various observational datasets, which can roughly represent the observational uncertainty (Fig.7 in the revised paper). Related statements have been added in Lines 322–332 on Page 11, or see the following sentences for convenience:

" Besides, a red horizontal bar is shown in Fig.7 centered at the 'REF' point on X-axis, the length of which can represent the observational uncertainty. Here, we use the area-weighted mean of standard deviations ($M_{SD}$) derived from multiple observations as the estimation of the observational uncertainty:

$$M_{SD} = \frac{\sum_{i=1}^{M}\sum_{j=1}^{N} w_j \cdot SD_{ij}^*}{M \cdot N}, SD_{ij}^* = \frac{SD_{ij}^{obs}}{SD_{ij}^t}$$

(2(

where $j$ ($i$) represents the grid (variable) index and $w_j$ is the area weighting. $SD_{ij}^{obs}$ is the standard deviation of multiple observations, which is calculated with the climatologies of REA1 and REA2 (Fig.7). Clearly, more observational data are desirable to derive a statistically meaningful standard deviation. Here, we only aim to illustrate how to show observational uncertainty in the VFE diagram. $SD_{ij}^t$ represents the inter-annual standard deviation of the reference, which is derived from the 40-year time series in autumn from 1961 to 2000. $M_{SD}$ is illustrated with the red bar in Fig.7 and it summarizes the mean dispersion of multiple observations in all grids for $M$ variables, which can roughly represent the overall uncertainty of observations."

*A weighting factor F has been introduced but is not discussed. In l. 430, the authors state that "The factor F in cMISS and uMISS is 2". What is the reasoning for this choice? What is recommended to users wanting to apply the MVIETool? Maybe give some examples for specific applications.*

**Response:**

Thanks for the valuable comment. We add a brief discussion about the choice of factor $F$ in the revised manuscript (Lines 112–120 on Page 4), which is also attached as follows:

" In terms of climate model evaluation, the pattern similarity is usually more important than the amplitude, because without pattern similarity, the accuracy of amplitude simulation is often less meaningful. Thus, one can set $F$ to be a value greater than 1 in Eq. (10) for general model evaluation purpose. In this case, MISS/MIEI is more sensitive to the change in the pattern similarity than the amplitude. Considering that MIEI has a geometric meaning when $F$ is 2, which represents the length of line segment CG (referring to Figure 3 in Xu et al., 2017). Thus, 2 appears to be a reasonable value of $F$ for general model evaluation purpose. Users can also change $F$ based on the application. For example, one may use a smaller $F$, say $F$=0.5, to give more weight to the amplitude if one wants to evaluate model ability to simulate the long-term trend of the multiple variables, e.g. the surface air temperature and specific humidity. In this case, one may have more concern about the values of the trends than their spatial patterns. "

*How is the grid cell area calculated that is used as weighting factor? Again, the statement (l. 189/190) that "If users want to consider area weighting in the statistics, the variables should be saved with the coordinate information (e.g., time, latitude, and longitude)" does not provide enough detail. How do the coordinates have to be defined? Is following the CF standard sufficient? Do the coordinates have to follow the CMOR conventions? Can area files ("fx" files in CMIP) be provided e.g. for irregular grids or is the analysis limited to regular grids?*

**Response:**

Thanks for the comment about the area weighting and coordinates of variables in MVIETool. A coordinate array represents the data coordinates for each index in the named dimension and should be *defined as a monotonic 1-D array* without missing values (referring to http://www.ncl.ucar.edu/Document/Manuals/Ref_Manual/NclVariables.shtml#CoordinateSubscripts). When coordinates assigned to the evaluated data, MVIETool can extract specified part of data according to the arguments for selecting coordinates' range (i.e., `Range_geo` and `Range_time` in Table 1) for evaluation. Meanwhile, only if the data are attached with the latitude coordinate, the tool can create area-weighting according to its value (referring to Eq. 19 in revised manuscript), which is used in the calculation of statistics. Besides, the tool can only deal with coordinates for regular gird at present, indicating that area-weighting are only valid for the evaluation in regular grid. User Guide of MVIETool has included these clear illustrations about the coordinate variable together with examples. As to area weighting, we have added detailed explanations in Lines 202–205 on Page 7, which is also available as follows:

"Currently, the tool can only deal with area weighting for regular grids and area weighting is calculated by the formula as:

$$w_j = sin(lat_j + d_{lat}) - sin(lat_j - d_{lat}) \tag{19}$$

where $lat_j$ is the latitude in *jth* grid and $d_{lat}$ is the difference in latitude between two adjacent zonal grids."

It is also not clear to me how time series of variables are handled. For example, additional information on (possibly) selecting a user specified time range is needed. Are attributes of the time coordinate such as calendar taken into account when calculating time means (e.g. number of days per month)? Is temporal interpolation done if the time resolution of two datasets does not match?

**Response:**

Thanks for the comment. The MVIETool assumes all input variables on the same grid and with the same coordinate variables, including time, latitude and longitude grid, and does not make further calculation to the datasets such as time mean and zonal mean before evaluation. As to the selection of time range in NCL version, the tool provides two options: First, one can specify the argument 'Range_time' with strings in the format: "YYYYMM", "YYYYMMDD", or "YYYYMMDDHH", where YYYY is the year, MM is the month, DD is the day, and HH is the hour, e.g., Range_time=(/"198101","199012"/). In this case, the coordinate variable assigned to the time dimension of input variables should additionally have a "calendar" attribution (referring to *http://www.ncl.ucar.edu/Document/Functions/Built-in/cd_calendar.shtml*). Second, one can specify the 'Range_time' with the values in time coordinate variable, e.g., Range_time= (/1,10/). We have illustrated how to set time coordinate in NCL as well as Python3 versions in User Guide.

*I was missing an overview (e.g. a table) on which models, experiments, years, time resolution, etc. of the model data and which reanalysis datasets have been used to create the example figures. This makes it impossible to reproduce the examples as an independent check (i.e. downloading the data yourself, applying the preprocessing steps and running the MVIETool) that the software is working as expected.*

**Response:**

Thanks for the comment. The CMIP5 datasets we used in examples are monthly mean derived from the first ensemble run of historical experiments during the period from 1961 to 2000 (Line262 on Page 9). We have presented the model name, institution and horizontal resolution of 10 CMIP5 models in Table A2 in Appendix of the revised paper, or see Table R1 for convenience.

*I have the impression that the MVIETool would benefit substantially from taking advantage of the infrastructure of existing model evaluation tools such as, for instance, the ESMValTool (Righi et al., Geosci. Model Dev., 2020). Such tools provide the possibility to preprocess all datasets in a consistent way regarding checking of input data, horizontal and vertical regridding, masking, time selection, vertical level selection, etc. I would like to encourage the authors to add some discussion on such a step as a possible outlook to the summary section.*

*It becomes increasingly more important to provide traceable and reproducible results. For model evaluation, this usually means providing a provenance record of the input data, used software, configuration, processing steps, etc. Is anything like this planned for the MVIETool? Again, I feel that the MVIETool could strongly benefit from taking advantage of the infrastructure of existing model evaluation tools that are already capable of providing provenance records.*

*Are there plans to continue development of MVIETool? I would recommend to add an outlook and thoughts about possible future directions to the summary section.*

**Response:**

Thanks for the reviewer's comments. We discussed potential future developments of the MVIETool in the revised manuscript (Lines 361–376 on Page 12–13), which is listed as follows for the reviewer's convenience:

"To make the evaluation methods available to more users, we also develop the MVIETool with Python3. Currently, the MVIETool 1.0 only provides some basic function to calculate statistics and generate figures for MVIE. We will continue to develop the tool to support more comprehensive evaluation. For example, the area weighting is only valid for the regular grid in MVIETool 1.0. In terms of irregular grids, the area weight can be derived from an additional data file that contains the grid area of each grid. To address observation uncertainty, the tool compares each individual observation against the average of multiple observations and the spread across various observations is taken as a measure of observational uncertainty. Another approach is to calculate the standard deviation of multiple observations as uncertainty estimation at present, which is also very basic. It warrants further investigation to develop a more sophisticated method that can estimate the impacts of observational uncertainty on model evaluation. In addition, no significance test is

available yet for difference between two vectors fields as well as the multivariable statistics, which also warrants for development in the future.

Furthermore, the Earth System Model Evaluation Tool (ESMValTool; Eyring et al., 2016; Weigel et al., 2020) is a systemic and efficient tool for model evaluation, which has been widely used in related studies (e.g., Valdes et al., 2017; Righi et al., 2020; Waliser et al., 2020). It has many distinct advantages, such as providing the well-documented analysis and no need for preprocessing of evaluated datasets, compared with our tool. In the follow-up work, we would not only devote to making advance in the function of MVIETool, but also intend to collaborate with the ESMValTool to include our package into it. In this way, users can benefit from the MVIETool with more convenience."

*L. 16: "MIEI" has not been defined yet*

**Response:**

We have defined MIEI in Line 93 on Page 4, or see the following sentence for convenience:

"$\text{MIEI}^2 = \frac{1}{M} \sum_{m=1}^{M} (L_{Am}^* - 1)^2 + 2 \cdot (1 - VSC)$       (7a) "

*L. 44: "most previous model performance metrics did not consider spatial weight": while this statement is true for the three examples mentioned, this is not the case for many other metrics and therefore needs rephrasing*

**Response:**

Thanks for the comment. We have revised the sentence as "The statistical metrics employed in Xu et al., (2016; 2017) did not consider spatial weight" in Lines 44–45 on Page 2 of the revised manuscript.

*L. 65: "... by dividing the corresponding rms value of the observation ...": it is not entirely clear to me what is divided by what, please consider rephrasing; if the original variable is divided by the rms of the observations, are all data on the same spatial and temporal grid? If not, what are the effects of this? What is the effect of averaging possibly available multiple observational datasets (see also general comments)?*

**Response:**

Thanks for the comment. MVIETool requires all input data on the same spatial and temporal grid. The sentence *"... by dividing the corresponding rms value of the observation ..."* here is unclear and we have rephrased the relative sentences as "We need to normalize each modeled variable by dividing the rms value of the corresponding observed variable. The normalized *M* variables are dimensionless and can be grouped into *M*-dimensional vector fields for model $A$ and observation $O$:" (Lines 65–67 on Page 3).

*L. 188: "variable" →"variables"*

**Response:**

We have replaced "*variable*" with "variables" in Line 199 on Page 7 of the revised manuscript.

*L. 196/197: What is meant by "put in parenthesis"? Where does this have to be done (source code, namelist, etc.)? Please be more specific.*

**Response:**

The MVIETool can evaluate scalar and/or vector fields. If the variables' names in the argument '`Varname`' (Table 1 in the manuscript) are put in parenthesis separated by comma such as (ua, va), the tool would treat it as a vector. This is explained in the revised manuscript " The MVIETool allows treating u850 (u200) and v850 (v200) as an individual vector field rather than two scalar fields. To declare a vector field, users can simply put the components of a vector in parenthesis separated by comma, e.g., (u850, v850) and (u200, v200) in the argument `Varname` of the tool (Table 1)." (Lines 210–212 on Page 7).

*L. 233: "one piece of observational data"→ e.g. "one observational dataset"?*

**Response:**

We have replaced "*one piece of observational data*" with "one observational dataset" in Line 249 on Page 9 of the revised manuscript.

*L. 234: "written in a new NetCDF file" →"written to a new NetCDF file"*

**Response:**

We have replaced " *written in a new NetCDF file*" to " written to a new NetCDF file" in Line 251 on Page 9 of the revised manuscript.

*L.258/258: What is meant by "better model performance" and "worse model performance"? Better or worse relative compared to what? Please be more specific and if possible more quantitative.*

**Response:**

Thanks for the comment. "better model performance" and "worse model performance" are determined based on the rank of model performance metrics in one evaluation. The filled color represents the value of the model performance metrics. We have clarified the sentence as " The filled color of each grid cell represents the value of statistical metrics. Lighter color indicates the model statistics is closer to observation and vice versa. The corresponding color bars can be shown below the metric table such as Fig. 6." in Lines 276–278 on Page 10 of the revised manuscript.

*L. 272: "relative" →"compared"*

**Response:**

We have replaced "*relative to*" with "compared with" in Line 292 on Page 10 of the revised manuscript.

**Table**

Table R1. Model names, institution and horizontal resolution for 10 CMIP5 models (M1–M10) used in the paper.

|      | Model | Institution | Horizontal resolution |
|------|-------|-------------|-----------------------|
| *M1* | BNU-ESM | College of Global Change and Earth System Science, Beijing Normal University(China) | 2.81° × 2.81° |
| *M2* | CCSM4 | NCAR (National Center for Atmospheric Research) Boulder(USA) | 1.25°× 0.94° |
| *M3* | CNRM-CM5 | Centre National de Recherches Meteorologiques / Centre Europeen de Recherche et Formation Avancees en Calcul Scientifique(France) | 1.41° × 1.41° |
| *M4* | BCC-CSM1-1 | Beijing Climate Center, China Meteorological Administration(China) | 2.81° × 2.81° |
| *M5* | FGOALS-g2 | LASG, Institute of Atmospheric Physics, Chinese Academy of Sciences; and CESS, Tsinghua University(China) | 2.81° × 3.05° |
| *M6* | GFDL-ESM2M | Geophysical Fluid Dynamics Laboratory(USA) | 2.5° × 2.0° |
| *M7* | GISS-E2-H | NASA Goddard Institute for Space Studies(USA) | 2.5° × 2.0° |
| *M8* | MIROC4h | Atmosphere and Ocean Research Institute (The University of Tokyo), National Institute for Environmental Studies, and Japan Agency for Marine-Earth Science and Technology(Japan) | 0. 56° × 0.56° |
| *M9* | MIROC-ESM-CHEM | Atmosphere and Ocean Research Institute (The University of Tokyo), National Institute for Environmental Studies, and Japan Agency for Marine-Earth Science and Technology(Japan) | 2.81° × 2.79° |
| *M10* | inmcm4 | Institute for Numerical Mathematics(Russia) | 2.0° × 1.5° |

---

## Author Response (AR2)

**Response Letter**

Dear Editor:

We sincerely thank you for your comments on our manuscript entitled "An improved multivariable integrated evaluation method and NCL code for multimodel intercomparison (MVIETool version 1.0)" (gmd-2020-310). We have revised our manuscript based on your comments. The changes are marked in red in the revised manuscript. Our point-by-point responses to your comments are as follows:

*I carefully evaluated your response to the reviewer comments and your revised manuscript. While your efforts in addressing the reviewer comments are very much appreciated, I think that your manuscript needs some further, minor revisions before publication:*

**Response:**

Thanks. We have revised the manuscript and related codes.

*First, I would like to encourage you to improve your manuscript in highlighting the novelty of your tool and the changes compared to previous versions. You provide a quite extensive discussion of the previous literature on performance metrics in the author's response, but your paper is rather short on this. I think the reviewer comment clearly shows that there is a need for clarification. So instead of addressing this issue only in the author's response, I suggest to extend the discussion of previous metrics and differences to your approach in the introduction.*

**Response:**

Thanks for the comment. We have added discussion about previous metrics and differences to our approach in the introduction of the revised manuscript (Lines 44–54 on Page 2), which is also listed as follows for the convenience:

"Moreover, the vector field statistics employed in Xu et al., (2016; 2017) did not consider area weighting, which is a limitation especially for an evaluation of the global field. Although area weighting was considered in many previous statistical metrics, e.g., correlation coefficient and standard deviation, they were used to evaluate scalar fields rather than vector fields (e.g., Watterson, 1996; Boer and

Lambert, 2001; Masson and Knutti, 2011). The consideration of area weighting in the definition of vector field statistics is one of the novelty of our study relative to previous studies (Taylor, 2001; Boer and Lambert, 2001; Gleckler et al., 2008; Xu et al., 2016; 2017). In addition, we also improve MVIE method to allow a mixed evaluation of scalar and vector fields. Furthermore, based on MIEI, a multivariable integrated skill score (MISS) for a climate model is proposed, which allows us to adjust the relative importance of different aspects of model performance. Finally, we develop a Multivariable Integrated Evaluation Tool (MVIETool version 1.0) to facilitate multimodel intercomparison. These efforts are expected to improve the accuracy and flexibility of the VFE and MVIE methods."

*Some technical corrections:*

*- L 44/45: consider rephrasing, spatial weight -> "area weighting"*

**Response:**

We have replaced " *spatial weight* " to "area weighting" in Lines 44–45 on Page 2 of the revised manuscript.

*- L65: This sentence still does not make sense and needs clarification: what is divided by what? Do you mean "... by dividing by the rms value of the corresponding observed variable"?*

**Response:**

Thank you for the comment. We have revised the sentence as "We need to normalize each modeled variable using the rms value of the corresponding observed variable." in Lines 69–70 on Page 3 of the revised manuscript.

*- L116: What does CG stand for?*

**Response:**

In the definition of MIEI (Xu et al., 2017), CG represents a line segment when $F$ is 2 (Fig. R1). The square root of the first term on the right side of Eq. 7a can be regarded as multivariable amplitudes' error, while the square root of the second term represents the pattern similarity error of multiple field. Xu et al. (2017) used them as the lengths of two perpendicular sides (i.e., BC and BG in Fig. R1) and constructed a

right triangle on VFE diagram. In this case, MIEI value can be regarded as the hypotenuse of the right triangle, which is line segment CG.

In the revised manuscript, the sentence has been reworded as "Considering that MIEI has a geometric meaning when $F$ is 2, which represents the length of a line segment (referring to line CG in Figure 3 in Xu et al., 2017)." in Lines 119–120 on Page 4 of the revised manuscript.

*- L 232/233: "No matter what kind of masks chosen, the missing points across all variables of one model are the same." This sentence needs clarification. Why would you apply the same mask for all variables of one model? Assume an observational temperature time series includes 10% missing data, while an observational ozone time series includes only 1% missing data: Would the tool discard valid data points from the ozone time series?*

**Response:**

Thanks for the constructive comment from the editor. The previous MVIETool unified missing points across all variables of one model, which is valid for the evaluation of the spatial field. However, this processing to missing points is not suitable for all evaluations, such as the situation proposed by the editor. To deal with more common situations, we have added a new argument —'Unif_VarMiss' (Table 1 in the manuscript) in the updated MVIETool. With the help of this argument, users can choose whether unify missing points across all variables of one model or not. We have also modified the sentence " *No matter what kind of masks chosen, the missing points across all variables of one model are the same.*" as "Further, whether to unify missing points across all variables of one model can also be chosen with the help of the argument Unify_VarMiss." in Lines 237–238 on Page 8 of the revised manuscript.

*- L276: "Lighter colors indicate that the model statistics is closer...."*

**Response:**

We have revised the sentence as "Lighter colors indicate the model statistics are closer to observation and vice versa" in Lines 281–282 on Page 10 of the revised manuscript.

*- L327: "... which is calculated using the climatologies..."*

**Response:**

We have revised the sentence as " $SD_{ij}^{obs}$ is the standard deviation of multiple observations, which is calculated using the climatologies of REA1 and REA2 (Fig. 7) " in Lines 331–332 on Page 11 of the revised manuscript.

*- Caption Fig. 7: "There is a red horizontal bar centered at the REF point of which length is as a measure of the observational uncertainty." Sentence needs rephrasing, e.g., "The observational uncertainty is indicated by the red horizontal line centered at the REF point." Same for the sentence in lines 322/323. And replace "red bar" by "red line".*

**Response:**

We have rephrased the sentence in the caption of Fig. 7 as " The observational uncertainty is indicated by the red horizontal line centered at the REF point.". In addition, we have replaced "*red bar* " with "red line" in Line 327/335 on Page 11 of the revised manuscript.

**Reference**

Xu, Z., Han, Y., and Fu, C.: Multivariable Integrated Evaluation of Model Performance with the Vector Field Evaluation Diagram, Geosci. Model Dev., 10, 3805–3820, 2017.

**Figure**

[Figure]

**Figure 3.** Schematic diagram displaying the relationship between the RMSVD, RMSD$_L$, and MIEI. The points A, B, and D represent different models. The RMSD$_L$ measures the overall difference between the modeled rms values and the observed ones. The line segment BC is vertical with respect to the VFE diagram. The length of line segment BG is determined based on the vector field similarity, which measures the overall pattern similarity of various scalar fields relative to the observed ones. Thus, the MIEI index takes both the pattern similarities and the rms values of various scalar fields into account.

**Figure R1.** Figure 3 in Xu et al. (2017)